# Vesicles driven by dynein and kinesin exhibit directional reversals without regulators

Ashwin I. D'Souza [1,5], Rahul Grover[1,5], Gina A. Monzon[1,2,5], Ludger Santen [2] ✉ & Stefan Diez [1,3,4] ✉

Intracellular vesicular transport along cytoskeletal filaments ensures targeted cargo delivery. Such transport is rarely unidirectional but rather bidirectional, with frequent directional reversals owing to the simultaneous presence of opposite-polarity motors. So far, it has been unclear whether such complex motility pattern results from the sole mechanical interplay between opposite-polarity motors or requires regulators. Here, we demonstrate that a minimal system, comprising purified Dynein-Dynactin-BICD2 (DDB) and kinesin-3 (KIF16B) attached to large unilamellar vesicles, faithfully reproduces in vivo cargo motility, including runs, pauses, and reversals. Remarkably, opposing motors do not affect vesicle velocity during runs. Our computational model reveals that the engagement of a small number of motors is pivotal for transitioning between runs and pauses. Taken together, our results suggest that motors bound to vesicular cargo transiently engage in a tug-of-war during pauses. Subsequently, stochastic motor attachment and detachment events can lead to directional reversals without the need for regulators.

Intracellular organelles such as endosomes, synaptic vesicles, and lipid droplets are transported as cargoes along polarized microtubule filaments by minus-end directed cytoplasmic dynein (referred to as "dynein") and plus-end directed kinesin motors. Multiple copies of dynein and kinesin are simultaneously present on individual cargoes leading to bidirectional motion[1–10]. This motion is characterized by fast runs in either direction and frequent directional reversals. The origin of these reversals and their regulation to achieve targeted transport remain poorly understood.

Directional reversals of cargoes might occur spontaneously or be induced by various mechanisms. The number and type of motors recruited to the cargo might regulate the relative abundance and spatial distribution of dynein and kinesin[6] or biochemical regulators might bias the transport direction by differentially supporting or impeding the teams of dynein and kinesin. With regard to the latter, cargo adapters (proteins that link motors to the cargo) that either activate dynein (e.g., BICD2[11,12], ninein[13]) or kinesin (e.g., nesprin-4[14])

have been identified as biasing factors. Additionally, some cargo adapters that function as scaffolds can simultaneously bind to dynein and kinesin and yet exclusively activate one motor over the other (e.g., HOOK3[15], TRAK2[16]). Moreover, patterns of microtubule-associated proteins (MAPs) at specific locations on the microtubule lattice can favor the passage of dynein over kinesin or vice versa[17–19]. Given these diverse mechanisms that can influence the behavior of motor-cargo systems in vivo, cell-free in vitro assays are essential to probe the interplay between dynein and kinesin on cargoes. Such assays typically involve examining the movement of dynein- and kinesin-bound cargoes on reconstituted microtubules. Native organelles extracted from cell or tissue lysates as cargoes exhibit many in vivo features of cargo transport, including directional reversals[1,4,6–9]. However, the role of regulators in inducing/controlling reversals cannot be ruled out as they might reside on the native cargo or cargo-bound motors (e.g., NudE-Lis1[20]) or be present in the cytosol if added to the assay. Therefore, to understand the role of different components in the transport

[1]B CUBE - Center for Molecular Bioengineering, TU Dresden, Dresden, Germany. [2]Center for Biophysics, Department of Physics, Saarland University, Saarbrücken, Germany. [3]Cluster of Excellence Physics of Life, TU Dresden, Dresden, Germany. [4]Max Planck Institute of Molecular Cell Biology and Genetics, Dresden, Germany. [5]These authors contributed equally: Ashwin I. D'Souza, Rahul Grover, Gina A. Monzon. ✉e-mail: l.santen@mx.uni-saarland.de; stefan.diez@tu-dresden.de

machinery and to determine whether the presence of dynein and kinesin on cargoes alone is sufficient to induce reversals, assays with synthetic cargoes and purified motors are necessary.

Recently, many in vitro reconstitution experiments with artificial assemblies of dynein and kinesin linked to DNA origami chassis, and short stretches of double-stranded DNA or glass surfaces have been reported[21–26]. However, all of them were limited in their capability to recapitulate key features of intracellular motility, such as fast unidirectional transport and directional reversals. The assemblies either remained stationary or moved at markedly low velocities suggesting a persistent tug-of-war between dynein and kinesin. Fundamentally, tugs-of-war are indeed hypothesized to occur in vivo[1,3,6–9]. Spatial elongations of endosomes isolated from *Dictyostelium discoideum*[9] and transport-induced changes in the size of mitochondria in *Xenopus laevis* tadpole neurons[27] have been regarded as a signature of the underlying opposing forces. Moreover, theoretical models describing a simple mechanical tug-of-war between cargo-associated dynein and kinesin motors have been used to explain the bidirectional transport of organelles such as endosomes[9] and phagosomes[1,3,28]. However, these tugs-of-war are intermittent and last only for short periods (1–2 s) before fast unidirectional transport resumes.

The lack of fast transport and directional reversals with artificial assemblies in vitro suggested that recapitulating features of intracellular cargo motility required additional components, which, in turn, regulated the activities of dynein and kinesin[21]. Such regulation would prevent simultaneous force generation by the opposite polarity motors, thereby reducing the prospect of a persistent tug-of-war. However, one key difference between artificial assemblies and intracellular cargoes is the nature of the cargo itself. The artificial assemblies consist of rigidly coupled fixed motor compositions. Intracellular cargoes, conversely, are usually vesicular structures made of lipid membranes that allow for motor diffusion. Properties of lipid membranes, such as geometry and fluidity, can influence transport characteristics. For example, spherical vesicles driven by multiple myosin Va motors can move faster than single myosin Va motors[29,30], and the transport efficiency of multiple lipid-anchored kinesin-1 motors is reduced by membrane fluidity[31]. So far, it has been unclear whether replacing artificial assemblies with vesicular cargoes would recapitulate the bidirectional features of intracellular cargo motility.

Here, we develop a well-controlled in vitro reconstitution assay with purified dynein and kinesin-3 as motors as well as large unilamellar vesicles of defined phospholipid composition as cargo. We show that vesicles driven by dynein and kinesin-3 exhibit the features of intracellular cargo motility, namely fast minus- and plus-end directed runs, pauses, and directional reversals. We observe that the simultaneous presence of dynein and kinesin-3 does not affect the velocity of the vesicles during unidirectional runs but increases the frequency of pauses. Furthermore, directional reversals are often preceded by a tug-of-war which manifests as vesicle elongation. In agreement with numerical simulations, our results suggest that motors diffusively anchored on vesicles do not hinder each other significantly during runs but engage in a tug-of-war during pauses where stochastic fluctuations in the number of engaged motors can lead to directional reversals without the necessity of regulators.

## Results

### Purified dynein–dynactin–BICD2 complexes and KIF16B motors are processive

We assembled a toolkit comprising functional minus- and plus-end directed motors. For the minus-end directed motors, we purified native *Homo sapiens* dynein and dynactin complexes from Human Embryonic Kidney 293 (HEK293) cells as well as *M. musculus* bicaudal 2 (BICD2) truncated to the first 594 amino acids[32] (BICD2N594, BICD2N594-eGFP) from *E. coli*, as dynein activator (Fig. 1a). The dynein–dynactin-BICD2N594-eGFP (DDB-eGFP; 1:2:1.5) complex was

active and moved along surface-immobilized microtubules (Fig. 1b, Supplementary Movie 1) with a median instantaneous (frame-to-frame) velocity of $-1.46 \pm 1.63\ \mu m\ s^{-1}$ (±IQR, Fig. 1c) and a median run length of $3.29 \pm 4.43\ \mu m$ (±IQR, Supplementary Fig. 1a). For the plus-end motor, we purified full-length *H. sapiens* KIF16B (with and without a C-terminal eGFP tag, Fig. 1a) from *Spodoptera frugiperda* (Sf9+) cells. KIF16B is a kinesin-3 motor responsible for the anterograde motility of early endosomes[33,34]. It contains a Phox homology (PX) domain at its C-terminus, which can directly bind to membranes containing phosphatidylinositol-3-phosphate (PI3P)[34,35]. KIF16B-eGFP motors exhibited processive motility along surface-immobilized microtubules (Fig. 1b, Supplementary Movie 2) with a median velocity of $0.80 \pm 0.63\ \mu m\ s^{-1}$ (±IQR, Fig. 1c) and a median run length of $0.63 \pm 0.68\ \mu m$ (±IQR, Supplementary Fig. 1b). KIF16B, along with other members of the kinesin-3 family, is hypothesized to largely exist as autoinhibited monomers in cells and undergoes dimerization only when recruited to cargoes[36]. However, our recombinantly expressed KIF16B was active and showed processive motility, indicating that at least a subset of motors can dimerize without a cargo.

### DDB–KIF16B–vesicles exhibit directional reversals in vitro

To observe the motility of vesicles driven by either or both opposite-polarity motors along microtubules, we first prepared large unilamellar vesicles (diameter of $132.5 \pm 48.5$ nm) containing DGS-NTA(Ni) and phosphoinositol-3-phosphate (PI3P) to attach DDB (dynein–dynactin–BICD2N594-8xHis) and KIF16B, respectively (Methods, Supplementary Fig. 2, and Supplementary Table 1). We then recorded the motility of motor-bound vesicles, diluted in an imaging buffer, along polarity-marked microtubules[37] under a total internal reflection fluorescence (TIRF) microscope (Fig. 2a). We first characterized the unidirectional vesicle motion towards the minus-end with DDB. DDB–vesicles were prepared by incubating vesicles with saturating amounts (7-fold higher than DGS–NTA(Ni) concentration) of BICD2N594-8xHis followed by the addition of pre-incubated dynein–dynactin (1:2) complexes (38 nM DDB). DDB–vesicles moved over long (>10 μm) distances towards the minus-end (Fig. 2b, Supplementary Movie 3). At this concentration of motors, DDB–vesicles often traversed the entire length of the microtubule. However, vesicle motility was frequently interrupted by pauses leading to a peak around zero in the histogram of instantaneous velocities and yielded a median velocity of $-0.40 \pm 0.94\ \mu m\ s^{-1}$ (±IQR, Fig. 2b).

We then characterized the unidirectional vesicle motion towards the plus-end with KIF16B. KIF16B–vesicles were prepared by incubating vesicles with 25 nM KIF16B. KIF16B–vesicles also exhibited robust motility over long distances (>10 μm; Fig. 2c, Supplementary Movie 4), often reaching the end of the microtubule. Processive runs of KIF16B–vesicles were also interrupted by pauses leading to a peak around zero in the histogram of instantaneous velocities and yielded a median velocity of $0.33 \pm 0.41\ \mu m\ s^{-1}$ (±IQR, Fig. 2c).

Next, we tested if vesicles would still move unidirectionally even in the presence of both DDB and KIF16B (dual-motor vesicle assay). DDB–KIF16B–vesicles were prepared by first incubating vesicles with BICD2N594 (7-fold in excess of DGS-NTA(Ni)) and 25 nM KIF16B, followed by the addition of dynein–dynactin (1:2) complexes (38 nM). We observed unidirectional vesicles either moving toward the minus-end or the plus-end, as well as vesicles exhibiting directional reversals from minus-end directed motion to plus-end directed motion or vice versa (Fig. 2d, magenta arrowheads, Supplementary Movie 5). As in the case of DDB–vesicles and KIF16B–vesicles, directed runs of DDB–KIF16B–vesicles were often interrupted by pauses. This pausing behavior appeared to be enhanced in the case of DDB–KIF16B–vesicles as indicated by a major peak around zero in the histogram of instantaneous velocities and yielded a median velocity of $0.01 \pm 0.41\ \mu m\ s^{-1}$ (±IQR, Fig. 2d). The pattern of

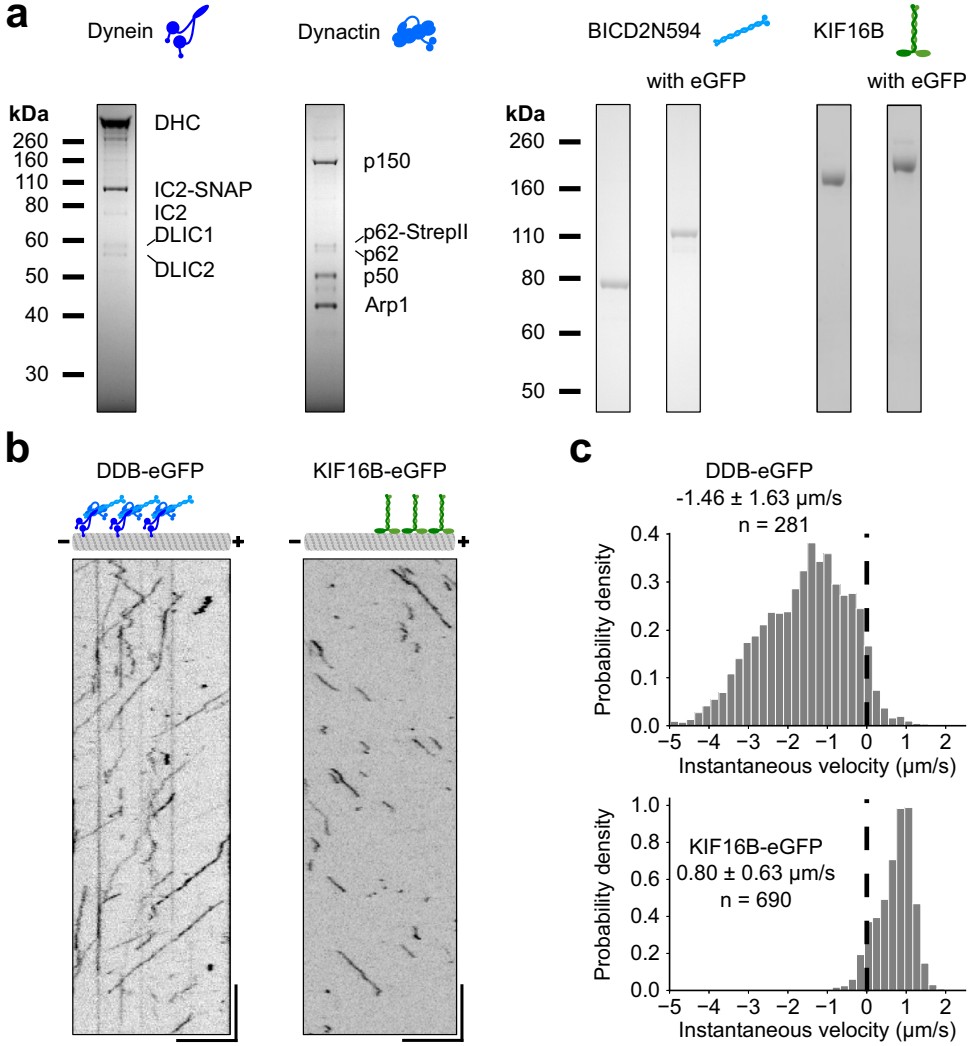

**Fig. 1 | Purified Dynein–Dynactin–BICD2N594 (DDB) complexes and KIF16B motors are processive. a** A representative SDS-polyacrylamide gel electrophoresis (SDS-PAGE) of purified Dynein, Dynactin, and BICD2N (truncated to first 594 aa., untagged and tagged with an enhanced green fluorescent protein (eGFP)) and Kinesin-3 (KIF16B, untagged and tagged with eGFP). Identity of DHC, IC2, p150, p62, p50, and Arp1 were confirmed with Western blotting using appropriate antibodies (see Methods). All experiments described in the paper were performed with at least two independent protein preparations. **b** Representative kymograph of single DDB complexes visualized with BICD2N-594-eGFP (DDB-eGFP, left panel) and single KIF16B-eGFP molecules (right panel). Scale bars: vertical 5 s, horizontal 5 μm. **c** Histograms of instantaneous velocity (velocity between consecutive frames) of single DDB complexes (upper panel) and single KIF16B-eGFP molecules (lower panel). Data are reported as median ± interquartile range (IQR). *n* represents the number of single molecules/complexes. The probability density is defined as the number of counts per bin divided by the product of total count and bin width. The integral over the histogram is equal to one. Source data are provided as a Source Data file.

motility observed with DDB–KIF16B–vesicles (fast unidirectional runs in either direction as well as occasional pauses and directional reversals) closely resembles the behavior observed with native cargoes such as early endosomes[3,6,7,9] and (auto) phagosomes[1,4,6,8,28,38], vesicles containing amyloid-β precursor protein[39], and synaptic vesicles[40] in vivo. Therefore, we conclude that we successfully reconstituted vesicle motility which mimics the transport of intracellular cargoes by multiple opposite-polarity motors, without the aid of any regulators.

Directional reversals of DDB–KIF16B–vesicles are an outcome of the simultaneous presence and activity of DDB and KIF16B on individual vesicles. However, unidirectional runs in either direction can be a result of either: (i) opposite-polarity motors not being present on the vesicles at the same time (i.e., KIF16B not present during minus-end runs and DDB not present during plus-end runs) or (ii) opposite-polarity motors being present at the same time but not hindering the driving motors. To distinguish between these scenarios, we performed dual-color motility assays where vesicles fluorescently labeled by Atto647N or Atto488 were incubated with either unlabeled DDB and eGFP-KIF16B (Supplementary Fig. 3a) or AlexaFluor647-DDB (DDB-647) and unlabeled KIF16B (Supplementary Fig. 3b), respectively. With unlabeled DDB and eGFP-KIF16B, we observed fluorescent signals from KIF16B-eGFP on all vesicles moving toward the plus-end. At the same time, we also observed the KIF16-eGFP signal on 76.2% of the vesicles (48 of 63) moving towards the minus-end (Supplementary Fig. 3a, blue arrowheads) and on all vesicles exhibiting reversals (Supplementary Fig. 3a, magenta arrowheads). Similarly, with DDB-647 and unlabeled KIF16B, we observed fluorescence from DDB-647 on 90% (36 of 40) of the vesicles moving toward the minus-end. Simultaneously, we observed DDB-647 on 88.5% of the vesicles (54 of 61) moving toward the plus-end (Supplementary Fig. 3b, green arrowheads) and on 85% (17 of 20) vesicles exhibiting reversals (Supplementary Fig. 3b, magenta arrowheads). This indicates that, for the most part,

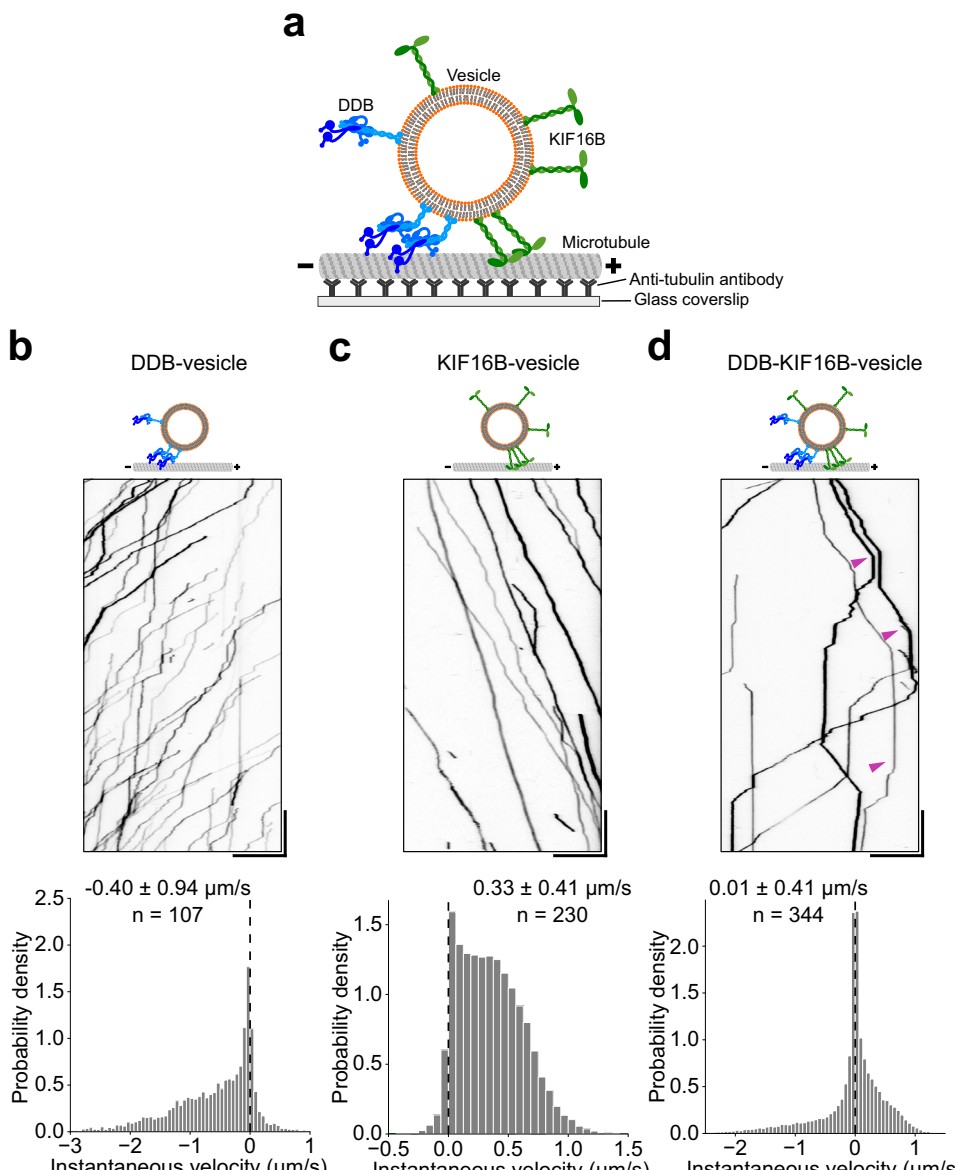

**Fig. 2 | DDB–KIF16B–vesicles exhibit directional reversals in vitro. a** Schematic diagram of vesicle motility assay. **b–d** Kymographs (upper panels) and velocity histograms (lower panels) of DDB–vesicle motility (**b**), KIF16B–vesicle motility (**c**), and DDB–KIF16B–vesicle motility (**d**) on polarity-marked microtubules. Magenta arrowheads mark vesicles that exhibit directional reversals. Scale bars: vertical 10 s, horizontal 10 μm. Numerical values are reported as median ± interquartile range (IQR). *n* represents the number of vesicles tracked from two independent experiments. Source data are provided as a Source Data file.

opposite-polarity motors are simultaneously present on the DDB–KIF16B–vesicles.

### Directionality of DDB–KIF16B–vesicles can be tuned by relative motor number

We tested if the transport direction of vesicles could be biased by simply tuning the relative abundance of DDB and KIF16B. To automate the determination of the transport direction of vesicles, we developed a segmentation algorithm that parsed the position-time tracks of vesicles into phases of negative runs, positive runs, and pauses (Fig. 3a). Tracks were first parsed by a change-point detection method that performs piecewise linear fitting to obtain optimized segments of constant velocity, followed by determining the slope of each segment to classify them as runs or pauses (Methods). Consequently, runs correspond to phases of directed transport, while pauses correspond to either phase of low velocity or diffusive movement with no significant net transport (Supplementary Fig. 3c). Based only on the

composition of the runs, individual tracks were classified as minus tracks (composed of only negative runs), plus tracks (composed of only positive runs), and reversal tracks (composed of at least one negative and one positive run). We reconstituted vesicle transport after incubation at a constant DDB concentration (38 nM) with varying KIF16B concentrations (10–75 nM) and determined the proportion of minus, plus and reversal tracks (Fig. 3b). The bulk motor concentration was used as a proxy for the number of vesicle-bound motors. DDB–vesicles (38 nM DDB) and KIF16B–vesicles (25 nM KIF16B) were used as controls. The frequency of minus tracks reduced while plus tracks increased with increasing KIF16B concentration. The inversion in the frequency of plus and minus tracks between 10 nM and 75 nM KIF16B shows that the direction of unidirectional motility is sensitive to this concentration range. The frequency of reversal tracks peaked at 25 nM KIF16B (30.2%). The fraction of negative and positive runs (normalized by the total distance traveled) also showed a similar trend: negative runs decreased while positive runs increased with increasing

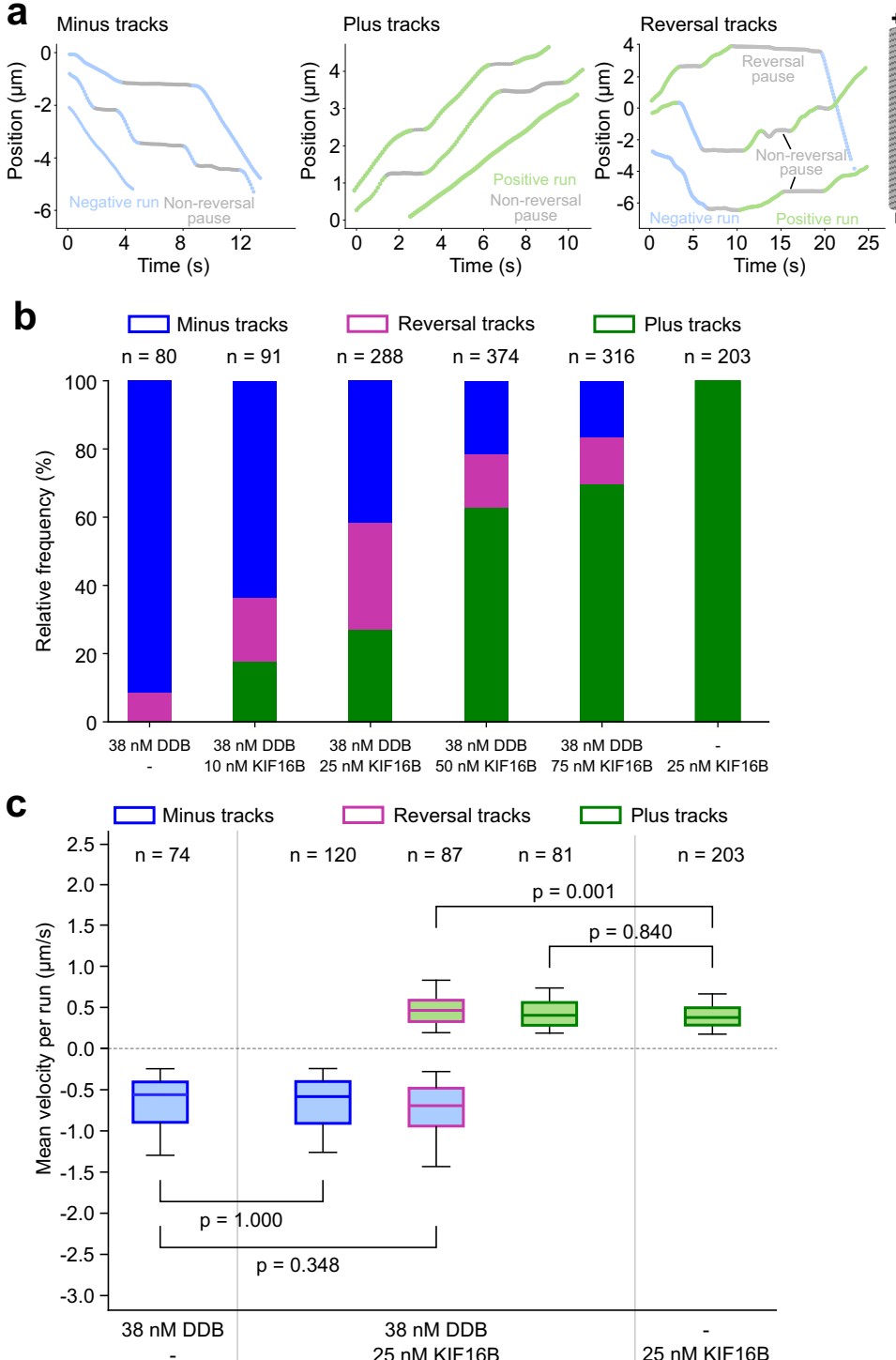

**Fig. 3 | Opposing motors do not affect the velocity of the driving motors.**
**a** Position-time tracks of vesicles that were segmented into runs and pauses using a segmentation algorithm based on the change-point detection method (see Methods). Tracks were classified as minus tracks when they consisted of only negative runs, plus tracks when they consisted of only positive runs, and reversal tracks when they consisted of at least one positive and one negative run. Pauses in between the runs were classified as non-reversal pauses when the run direction remained the same and reversal paused when the run direction reversed after the pause. **b** Proportion of minus (blue), reversal (magenta), and plus (green) tracks obtained from DDB–KIF16B–vesicles incubated with 38 nM DDB and various concentrations of KIF16B (10, 25, 50, 75 nM). DDB–vesicles and KIF16B–vesicles were used as controls. The small portion of reversal tracks identified for the DDB–vesicles are attributed to the partly diffusive nature of the DDB motors[24]. n represents the total number of tracked vesicles for a given condition (data pooled from two independent experiments). **c** Box plots of mean velocities of negative runs (blue filled boxes) and positive runs (green filled boxes) from minus (blue outline), plus (green outline), and reversal tracks (magenta outline) obtained from DDB–KIF16B–vesicles incubated with 38 nM DDB and 25 nM KIF16B. Negative velocities from minus tracks of DDB–vesicles (38 nM DDB) and positive velocities from plus tracks of KIF16B–vesicles (25 nM KIF16B) are shown as controls. Box plots indicate the median (middle line), 25th, 75th percentile (box), and 5th and 95th percentile (whiskers). n represents the number of tracked vesicles. *p*-Values were computed by comparing the respective samples using an independent two-tailed *t*-test with Bonferroni correction. Source data are provided as a Source Data file.

KIF16B concentration (Supplementary Fig. 3d). We conclude that vesicles exhibiting directional reversals can be biased to move unidirectionally by increasing the number of motors moving in that direction.

## Opposing motors do not affect the velocity of the driving motors

Although the presence of opposing motors was insufficient to induce reversals in the transport of all vesicles, it might still alter the characteristics of the runs. For example, microtubule binding and force generation by opposing motors might cause a reduction in the transport velocity of the driving motors. This kind of interference has been observed previously for artificial assemblies of dynein and kinesin[21,23,24] and microtubules gliding on a lawn of surface-bound dynein and kinesin[26]. Therefore, we asked if a similar reduction in velocity also occurs during phases of unidirectional runs for vesicles undergoing reversals. We addressed these possibilities by investigating (i) whether vesicles in our dual-motor vesicle assays were slower than vesicles in our single-motor vesicle assays and (ii) whether the reversing vesicles in our dual-motor vesicle assays were slower than vesicles exhibiting unidirectional motion only.

Towards this end, we focused on vesicles incubated with 38 nM DDB and 25 nM KIF16B as we observed the highest proportion of reversal tracks under this condition. When comparing the negative velocities (Fig. 3c, blue filled boxes; Supplementary Tables 2 and 3) of DDB−vesicles and dual-motor vesicles (from minus tracks and reversal tracks, blue and magenta outlines), we did not observe a considerable reduction in the velocity of the negative runs in the presence of KIF16B. Likewise, positive velocities (Fig. 3c, green filled boxes; Supplementary Tables 2 and 3) of KIF16B−vesicles and dual-motor vesicles (from plus tracks and reversal tracks, green and magenta outlines) were not significantly different. Further supporting these findings, the velocity histogram of dual-motor vesicles, especially the negative and positive tails, resembled a combination of the velocity histograms of DDB−vesicles and KIF16B−vesicles (Supplementary Fig. 3e). Similar results were observed for vesicles incubated with 38 nM DDB and 10, 50, and 75 nM KIF16B (Supplementary Fig. 3f, Supplementary Tables 2 and 3). As expected, the velocity during the pauses was significantly lower than during the runs (Supplementary Fig. 4). Therefore, we conclude that DDB and KIF16B, as opposing motors, do not functionally interfere with the activity of the driving motors during unidirectional runs.

## DDB and KIF16B engage in a tug-of-war during vesicle pausing

While we did not observe a slowdown of vesicles in the dual-motor vesicle assay during unidirectional runs, pausing became the dominant feature in the motility of DDB−KIF16B−vesicles (see major peak around zero in the histogram of instantaneous velocities in Fig. 2d). These pauses might indicate periods where DDB and KIF16B simultaneously generate force against each other, i.e., engage in a tug-of-war with no net movement. However, we also observed pauses in single-motor vesicle assays (DDB−vesicles and KIF16B−vesicles), suggesting that not all pauses resulted from a tug-of-war between DDB and KIF16B. Rather, at least part of the pauses could arise from a fraction of motors on the vesicles that are inactive (i.e., they interact with the microtubule in a stationary manner, as observed in Fig. 1b). Therefore, we investigated if the pauses of dual-motor vesicles differed from those of single-motor vesicles. Focusing again on vesicles incubated with 38 nM DDB and 25 nM KIF16B, we measured the spatial frequency of pauses (i.e., the number of pauses per micrometer of distance traveled) by calculating the mean of the ratio of the number of pauses to the distance traveled by individual vesicles. We weighted this ratio by the fraction of distance traveled by all vesicles of a given type (i.e., from either minus, plus, or reversal tracks; Methods) as there were many short tracks that either did not pause or reached the microtubule ends. Dual-motor vesicles exhibiting directional reversals had a higher spatial pause frequency ($0.67 \pm 0.05\,\mu m^{-1}$, weighted mean ± weighted standard

error of mean) than dual-motor vesicles moving unidirectionally ($0.26 \pm 0.03\,\mu m^{-1}$) and single-motor vesicles ($0.25 \pm 0.05\,\mu m^{-1}$ for DDB−vesicles and $0.31 \pm 0.03\,\mu m^{-1}$ for KIF16B−vesicles; Fig. 4a). We hypothesize that on vesicles exhibiting directional reversals stochastically the number of opposing motors was such that their counteracting forces were more balanced than on vesicles moving only unidirectionally. Thus vesicles exhibiting directional reversals have a higher probability of running into motor configurations which lead to pauses as compared to vesicles moving unidirectionally.

Directional reversals ($n = 145$) were either instantaneous (19%) or occurred after a pause (81%). However, pauses did not always result in a directional reversal (Fig. 3a). To investigate if pauses occurring before a directional reversal ("reversal" pauses) were different from pauses that resulted in vesicles continuing to move in the same direction ("non-reversal" pauses), we examined the duration of these two types of pauses. We found that the duration of reversal pauses (5.2 ± 14.5 s, median ± IQR) was longer than the duration of (i) non-reversal pauses (2.4 ± 4.4 s) in the reversal tracks of our dual-motor vesicle assays, (ii) pauses of DDB−vesicles (2.3 ± 2.5 s) and (iii) pauses of KIF16B−vesicles (1.8 ± 2.7 s, Fig. 4b). We interpret the longer durations of the reversal pauses to be an outcome of an extended tug-of-war between DDB and KIF16B, in line with previous reports[21–26]. Together, these data suggest that opposing motors can interrupt the motion of the driving motors, either causing the vesicle to reverse instantaneously or to pause. However, vesicle pausing is not indefinite, as the vesicle resumes transport in either the same or the opposite direction.

Occasionally, and mostly during pauses, we observed the pronounced elongation of larger vesicles (Fig. 4c, d; Supplementary Movie 6). Indicative of a tug-of-war between DDB and KIF16B, these elongations occurred in both directions. To determine if vesicle elongation correlated with the state of motion, we performed a dual-motor vesicle assay with some modifications. We used larger vesicles (144.9 ± 57.3 nm) to better observe and quantify changes in vesicle morphology and employed an alternative fitting model for particle tracking to estimate vesicle dimensions (see Methods and Supplementary Fig. 5a). We did observe that vesicle elongations were more pronounced during lower velocities (which include the pausing states) and less pronounced during higher velocities (Supplementary Fig. 5b, c).

## Low numbers of attached motors are critical to observe reversals

While previous experimental attempts to reconstitute bidirectional transport in vitro resulted in stationary cargoes due to a persistent tug-of-war between dynein and kinesin, computational models have recapitulated unidirectional runs and reversals[9,41,42]. To identify parameters that are key to obtaining reversals and to obtain insight into the dynamics that cause and eventually resolve a tug-of-war, we developed a stochastic stepping model (Fig. 5a, Methods). Briefly, DDB and KIF16B are modeled as Hookean springs on a spherical cargo. As given in the experiments, the motors are a mixture of active, inactive and diffusive motors. Our cargo surface is divided into a small attachment area[6,43] and a large reservoir. Motors in the attachment area can attach to the microtubule and generate force. Upon detachment from the microtubule, motors are instantaneously exchanged for new motors (either active, inactive or diffusive; with an individual force-free stepping rate) of the same type (DDB or KIF16B) from the reservoir into the attachment area (see Supplementary Tables 4 and 5 for the parameters used). This way, we account for the diffusion of motors within the vesicle membrane. The model with either DDB or KIF16B alone can recapitulate the experimentally observed instantaneous velocity histograms of single-motor vesicles (Supplementary Fig. 6a, b).

We simulated tracks of cargoes with a constant mean number of DDB motors ($N_D = 8$) and varying mean numbers of KIF16B motors ($N_K = 6, 14, 20, 25$) in the attachment area (Methods). As described

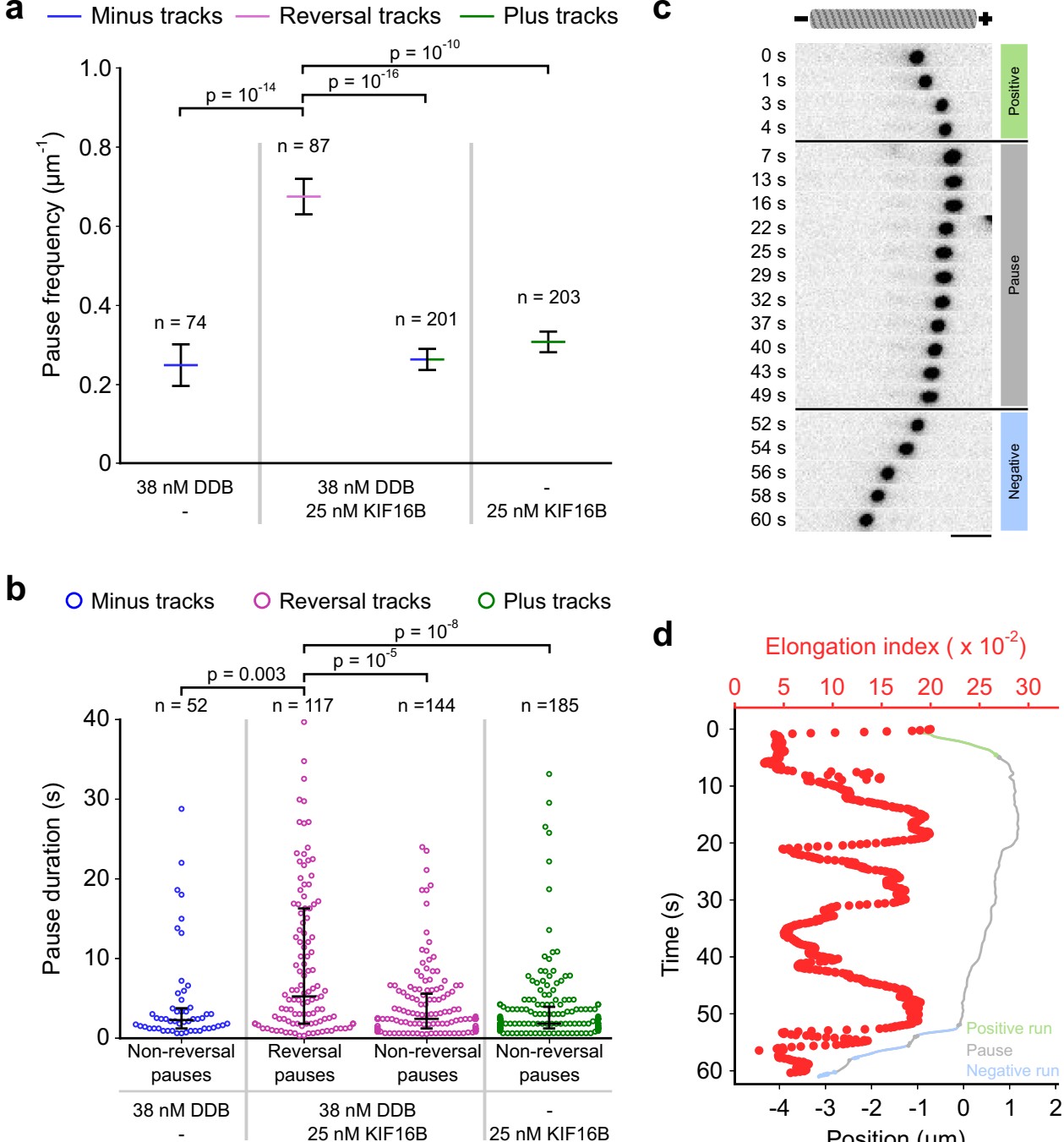

**Fig. 4 | DDB and KIF16B engage in a tug-of-war during vesicle pausing. a** Plot of spatial pause frequency (the number of pauses divided by the distance traveled per track) weighted by the proportion of total distance traveled by vesicles of a given type (minus, plus, or reversal tracks) in single- and dual-motor vesicle assays. Spatial pause frequencies of minus and plus tracks from dual-motor vesicles were combined. The central lines mark the weighted means and the error bars represent the weighted standard error of the mean (see Methods). *n* represents the number of tracks analyzed in each condition. *p*-Values were obtained from weighted two-sample, two-tailed Kolmogorov–Smirnov tests with Bonferroni correction. **b** Beeswarm plots of pause durations of minus tracks from DDB–vesicles, plus tracks from KIF16B–vesicles, and reversal tracks from DDB–KIF16B–vesicles. The central lines mark the median durations while the whiskers span the 25–75 percentile values. *p*-Values were computed by pairwise two-tailed Mann–Whitney *U* tests with Bonferroni correction. *n* represents the number of pauses for a given condition. **c** Timelapse images of a large vesicle undergoing an elongation along the long axis of a microtubule (not shown). Vesicle elongation coincides with a paused state (7–49 s, where the velocity is slower than 0.2 µm/s while the vesicle is more spherical again during the fast runs (plus-end directed at 0–4 s and minus-end directed 52–60 s). Scale bar: 2 µm. **d** Position-time trace of the vesicle from (**c**) segmented into runs and pauses and overlaid with the time-course of elongation index. The elongation index measures the shape of the vesicle and is calculated as the ratio of the difference to the sum of the long and short axis of the vesicle. Higher elongation indices indicate elongated vesicles. The "pulsing" in the elongation index during the pausing state occurs due to consecutive elongation and relaxation of the vesicle, presumably due to multiple tug-of-war events between DDB and KIF16B (see Supplementary Fig. 5 for more examples and further analysis). Source data are provided as a Source Data file.

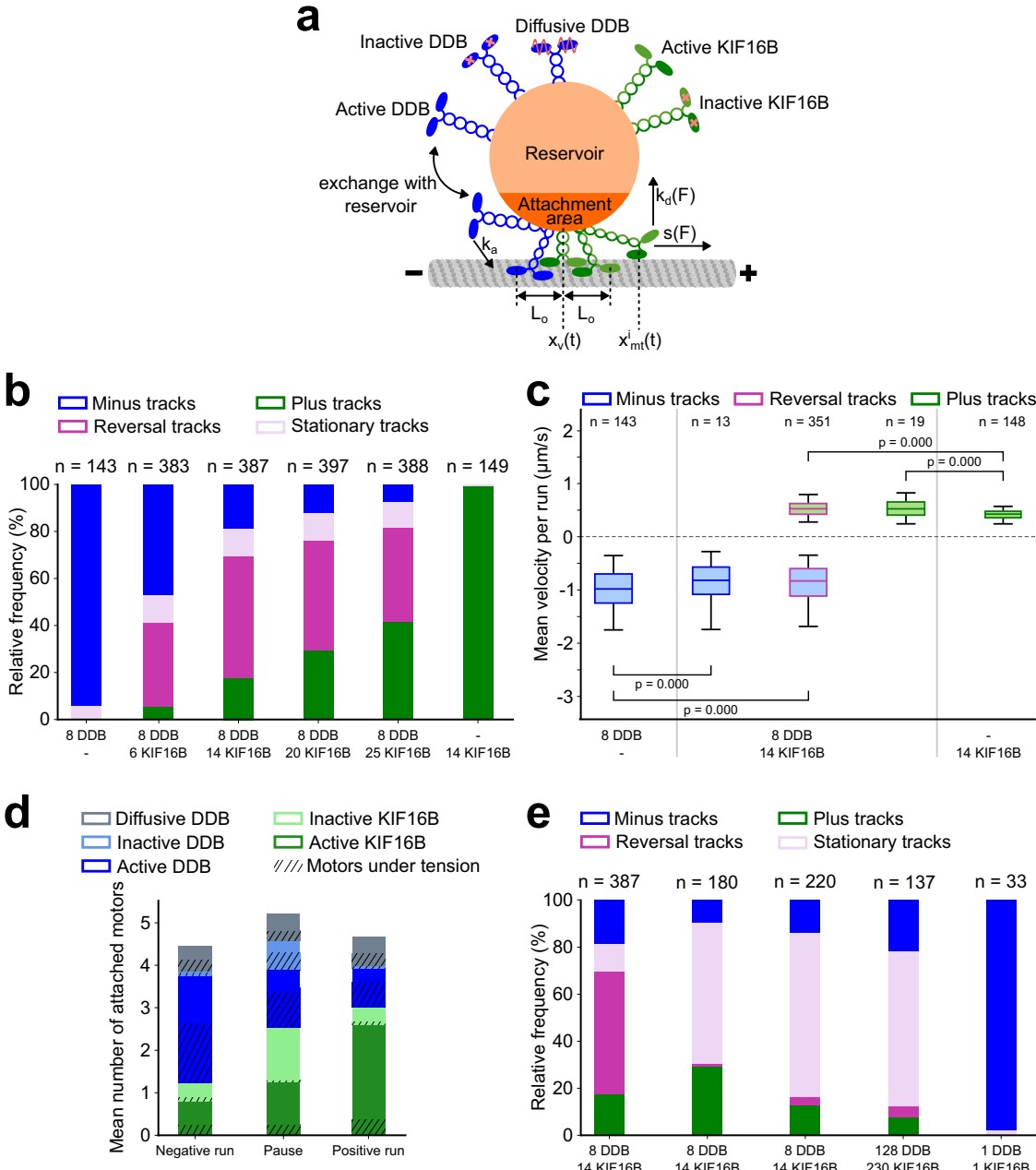

**Fig. 5 | Low numbers of attached motors are critical to observe reversals.**
**a** Schematic of the model. Motors bound to the cargo comprise active KIF16B, inactive KIF16B, active DDB, inactive DDB, and diffusive DDB. Cargo is divided into an attachment area and a reservoir. Motors in the attachment area can attach the microtubule with a constant attachment rate $k_a$ and detach with force-dependent detachment rates $k_d(F)$. Active DDB and active KIF16B motors, step with force-dependent stepping rates $s(F)$. Attached, diffusive DDB motors diffuse in the harmonic potential of the springs. Motors stretched beyond $L_{DDB/KIF16B}$ are under tension and exert a force on the cargo ($|F| > 0$). **b** Proportions of minus (blue), plus (green), reversal (magenta), and stationary (lilac) tracks obtained from simulations of cargoes with 8 DDB motors and 6, 14, 20, and 25 KIF16B motors in the attachment area; cargoes with either 8 DDB or 14 KIF16B only are controls. $n$ represents the total number of simulated tracks for a given condition, and plotted data was derived from tracks that were resampled to resemble the experimental distribution of track durations (Methods). **c** Box plots of mean velocities of negative runs (blue filled boxes) and positive runs (green filled boxes) from minus (blue outline), plus (green outline), and reversal tracks (magenta outline) obtained from simulations of

cargoes with 8 DDB and 14 KIF16B motors in the attachment area; negative velocities from minus tracks of DDB–vesicles and positive velocities from plus tracks of KIF16B–vesicles (8 and 14 motors respectively) are controls. Box plots indicate the median (middle line), 25th, 75th percentile (box), and 5th and 95th percentile (whiskers). $n$ represents the number of simulated cargoes. $p$-Values were computed from weighted two-sample, two-tailed Kolmogorov–Smirnov tests with Bonferroni correction to account for multiple comparisons. **d** Stacked bar plots of mean attached numbers of motors during negative runs, pauses, and positive runs from simulations of cargoes with 8 DDB and 14 KIF16B motors in the attachment area. Shaded areas represent motors under tension. Data from 396 simulated cargoes were used to construct these plots. **e** Proportion of minus (blue), plus (green), reversal (magenta), and stationary (lilac) tracks obtained from simulations of cargoes with (i) 8 DDB and 14 KIF16B motors in the attachment area, (ii) with 32-fold higher attachment rate each, (iii) with 20-fold lower detachment rate each, (iv) 128 DDB motors and 230 KIF16B motors, and (v) one active DDB motor competing against one active KIF16B motor. Source data are provided as a Source Data file.

before, we segmented these tracks into runs and pauses (Supplementary Fig. 7a). Similar to our experimental results from dual-motor vesicles, we obtain minus, plus and reversal tracks, and the proportion of plus tracks again increases upon increasing the number of KIF16B motors (Fig. 5b, compare to Fig. 3b). We also identify a small number of stationary tracks where the cargo is not moving at all. Such tracks were also present in our experimental results but were not included in the evaluation because we could not rule out that they occurred at microtubule junctions or microtubule ends. In agreement with our experiments, velocities of unidirectional runs are not significantly affected by the presence of opposing motors (Fig. 5c, Supplementary Fig. 7b, Supplementary Tables 6 and 7). The model did not recapitulate the experimentally observed prolonged durations of reversal pauses compared to non-reversal pauses (Supplementary Fig. 7c, Supplementary Table 8). This discrepancy suggests that in the experimental system, mechanisms which partially stabilize the tug-of-war configurations (e.g., by increasing the rebinding probability of detached motors) might be at play. However, in line with our experimental data, we do observe more instances of higher absolute forces during lower velocities (which include the pausing states) than during higher velocities (Supplementary Fig. 5b, c). Removing the inactive motors yields overall similar results (Supplementary Fig. 8a, b, Supplementary Table 9) except for the pauses becoming shorter and more diffusive (Supplementary Fig. 8c).

Importantly, our numerical simulations allow us to shed light on the configurations of the motors interacting with the microtubule during the runs and pauses (Fig. 5d, Supplementary Fig. 9). During pauses, we observe an increased number of attached inactive motors and comparable numbers of attached active KIF16B and active DDB motors. In contrast, unidirectional (positive and negative) runs are characterized by high numbers of active driving motors along with low numbers of inactive and opposing motors. Moreover, we observe that during runs and pauses, only a small fraction of the active attached KIF16B motors are under tension (generating force), with the rest forming a pool of untensioned motors. In contrast, a large fraction of the active attached DDB motors is tensioned (Fig. 5d). Counteracting forces are balanced by two distinct mechanisms: On one hand, tensioned KIF16B motors exhibit a high force-dependent detachment rate because only a few motors share the load. After detachment, however, attached untensioned KIF16B motors are readily available to come under tension and take over the load. On the other hand, tensioned DDB motors exhibit a low force-dependent detachment rate because many motors share the load. Together with their lower force-free detachment rate (compared to KIF16B), they resist the load longer before detachment.

Using our numerical simulations, we were interested in identifying the conditions that lead to directional reversals. In particular, it is known that the attachment/detachment kinetics and the number of motors can be critical to cargo transport, especially during mechanical competition between opposite-polarity motors[44,45]. For example, individual motors rigidly bound to cargo may rapidly reattach (i.e., significantly faster than out of solution) after detachment due to their immediate and retained proximity to the microtubule. However, motor diffusion in the membrane is expected to prevent such rapid reattachment in the case of vesicular cargo. We, therefore, tested if increasing the attachment rate of the motors would reduce the likelihood of directional reversals. We simulated cargo transport with 8 DDB and 14 KIF16B motors with increased attachment rates (32-fold higher for both DDB and KIF16B). We found that a large fraction of tracks then becomes stationary (Fig. 5e). Moreover, the number of attached motors is increased (Supplementary Fig. 10a) and the velocities of unidirectional runs in the presence of opposing motors are reduced (Supplementary Fig. 10b, Supplementary Tables 10 and 11). We can recapitulate the same effect by either reducing the detachment rate of the motors (20-fold lower for both DDB and KIF16B) or by

simply increasing the number of motors bound to the cargo (128 DDB and 230 KIF16B motors) (Fig. 5e). When modeling only one KIF16B motor competing against one DDB motor at high attachment rates, we observe exclusively stationary or unidirectional tracks (Fig. 5e).

## Discussion

In this study, we reconstituted the motility of synthetic vesicles bound to purified DDB motor complexes and KIF16B motors in vitro. We observed transport with similar features as vesicles moving in vivo. In particular, our assay recapitulated phases of unidirectional fast runs and pauses, as well as directional reversals. Velocities of unidirectional runs, both towards the minus- or the plus-end, were not significantly affected by the presence of the opposing motors. However, opposing motors interrupted vesicle transport by either causing instantaneous directional reversals, or vesicle pausing. Occasionally, vesicles elongated along the long axis of the microtubule before reversing their directions, indicating a tug-of-war between the opposite polarity motors.

Thus far, motility of only non-vesicular assemblies driven by dynein and kinesin (either as individual motors or ensembles) has been reconstituted. Neither fast unidirectional runs nor directional reversals could be recapitulated in those experiments; the studied assemblies rather exhibited slow motility or highly stable tugs-of-war, characterized by long stationary phases[21–24,26]. This suggested that dynein and kinesin cannot be active simultaneously on intracellular cargoes[22,42,46]. Consequently, dynein and kinesin activities on intracellular cargoes were proposed to be reversibly coordinated by regulators[10,39,47–50]. These regulators, either cytosolic or membrane-bound, were hypothesized to prevent tugs-of-war by reciprocally activating dynein and kinesin, thereby preventing cargoes from remaining stationary. However, we show that directional reversals do not require any regulators. Instead, stochastic fluctuations in the number of engaged dynein and kinesin motors are sufficient to induce cargo reversals (Fig. 6).

We performed stochastic numerical simulations to gain insights into the motor configurations during bidirectional cargo transport. We, and others, have previously reported models of bidirectional transport[26,45,51–56], some of which consider the diffusion of motors on cargo surfaces[53,54]. For example, Müller et al. 2008 first reported a modeling approach which shows bidirectional motion[56]. However, the study did not consider explicit motor positions and assumed that the motors of the same type share forces equally, contrasting different experimental setups[42,55,57]. Taking explicit motor positions into account, we find that typical runs are characterized by 2–3 driving active motors and the occasional attachment of opposing and inactive motors (mean number less than 1, Fig. 5d). To transition to a pause (characterized typically by a force balance between about one active motor of each kind, stabilized by one to two inactive motors of either type), one to two driving active motors need to detach and one opposing as well as one to two inactive motors need to attach. A new run in either direction can then be initiated by the detachment of one active motor or the attachment of one to two active motors (which pull off the inactive and opposing motors). Such changes in the configurations of attached motors occur due to stochastic attachment and force-dependent detachment events. Thus, particularly at low numbers of attached motors, single-motor detachment and attachment events can change the state of motility and lead to directional reversals. In agreement with this, directional reversals are suppressed in our simulations when the numbers of motors engaged with microtubules are high, which can be a result of (i) a high total number of motors, (ii) a high motor attachment rate, and (iii) low motor detachment rate (Fig. 5e).

In previous in vitro reconstitution studies[21–26], the rigidly coupled motors likely exhibited very high microtubule attachment rates due to their constant proximity to the microtubule. In fact, large attachment rates were required to computationally recapitulate the motility of

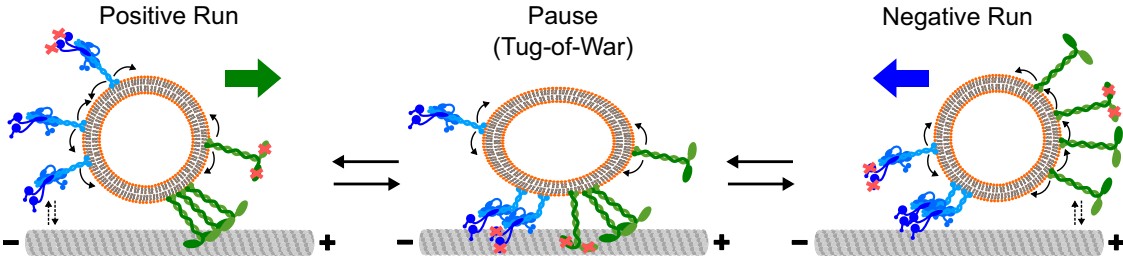

**Fig. 6 | Model for directional reversals of vesicles driven by opposite-polarity motors.** Active and inactive (with red cross) DDB and KIF16B motors can diffuse (curved arrows) within the vesicle membrane. A negative or a positive run is initiated by the attachment of active DDB or KIF16B motors to the microtubule, respectively. Only the driving team of motors is attached to the microtubule during the runs. Opposing motors occasionally attach to the microtubule but high load forces by the driving motors cause them to rapidly detach (dashed arrow pairs). Stochastic motor attachment/detachment events result in fluctuations in the number and type of engaged motors. These events include different permutations of (i) attachment of opposite-polarity motors, (ii) attachment of inactive motors of either type, or (iii) detachment of driving motors. Such fluctuations can result in a tug-of-war between the opposite-polarity motors, leading to vesicle pausing (with occasional elongation). Likewise, these fluctuations can cause the vesicle to transition from a pause to a run in either direction. A low number of engaged motors is critical for the fluctuations to cause these transitions and to result in directional reversals.

DNA-based scaffolds of dynein and kinesin[25]. We hypothesize that a high attachment rate, which also causes a depletion of the pool of unattached motors in the attachment area, is responsible for the lack of directional reversals in previous studies. In contrast, the rapid diffusion of detached motors away from the microtubule on the membrane of vesicular cargo leads to a reduced (re)attachment rate such that only a small number of motors is attached at any given time. Consequently, stochastic motor attachment, detachment events or both can readily change the motility state and lead to directional reversals. Slower motor diffusion on the cargo surface would lead to higher (re)attachment rates and, therefore, a reduction in directional reversals.

A key feature of our assay is the simultaneous but independent activities of the opposite-polarity motors. This was engineered by attaching DDB and KIF16B to the vesicles with orthogonal attachment schemes: (i) binding of KIF16B via its PX domain to PI3P lipids mimics the physiological scenario and (ii) binding dynein via truncated BICD2 and Ni-conjugated phospholipid (DGS−NTA(Ni)) emulates dynein recruitment to endogenous cargoes via full-length BICD2 and Rab6, where Rab6 relieves the autoinhibition of BICD2 upon binding[32]. However, the binding of dynein and kinesin to intracellular cargoes may be interdependent. Recent in vitro studies have shown that individual dynein or kinesin cargo adapters such as HOOK3[15] and TRAK2[16] can scaffold (and activate, in the case of TRAK2) both motors into a single dynein–kinesin–adapter complex. Remarkably, unlike previous DNA-based assemblies of single dynein and kinesin molecules[21,23,24], these complexes exhibited only fast unidirectional transport (either toward the minus- or the plus-end) indicating the exclusive activity of only one type of motor at any instance in a complex. As such, the recruitment of multiple dynein–kinesin–adapter complexes to cargo is functionally equivalent to independently attaching dynein and kinesin, as reconstituted in our experiments. The significance of such simultaneous recruitment to, but exclusive activity of dynein and kinesin on, intracellular cargoes remains to be determined.

How the overall directionality of different cellular cargoes is biased towards its indented destinations has yet to be well understood. In our in vitro assays, we observed a strong influence of relative motor concentrations on the transport direction of vesicles (Fig. 3b) which agrees with previous theoretical studies that have identified relative abundance of the motors as one of the key determinants of transport direction[9,26,41,42,58]. In addition, in vivo, it is expected that apart from the number and relative strengths of active kinesins vs. active dynein motors, the direction will be influenced by external factors such as adapters, MAPs, and tubulin posttranslational modifications, which can selectively alter the attachment, detachment and stepping rates of the opposite polarity motors. For example, landing of single molecules of DDB but not KIF1A (a kinesin-3 motor) was shown to be severely affected on MAP9-decorated microtubules[17]. Likewise, tau exerts differential effects on the processivities of dynein and kinesin[1,19]. This was hypothesized to be responsible for the enhanced minus-end motility of phagosomes on tau-decorated microtubules in vitro[1].

Taken together, the incorporation of lipid membranes in in vitro motility assays provides an exciting opportunity to study different facets of multi-motor transport, including—but not limited to—motor–membrane interactions[29–31,43,51] and motor–microtubule interactions[59]. In the future, it will be intriguing to extend the presented experimental and theoretical approach to systematically analyze the effects of MAPs, tubulin posttranslational modifications, cargo adapters and other regulators on the different features of intracellular cargo transport.

## Methods
### Reagents
Plasmids containing the gene for DIC2-SNAPf and p62-Halo were a kind gift from Prof. Samara Reck-Peterson, University of California San Diego[13]. Full length *H. sapiens* KIF16B gene in pFastBac vector was a kind gift from Prof. Marino Zerial, MPI-CBG, Dresden[34]. cDNA encoding the gene for *M. musculus* BICD2 was obtained from genomics-online (ABIN3826068). 18:1 (Δ9-Cis) DOPC (850375P), 18:1 (Δ9-Cis) DOPE (850725P), 18:1 PI3P (850150P), 18:1 DGS-NTA(Ni) (790404C) were obtained from Avanti-Polar while 18:1 DOPE-Atto647N (45-2247) was obtained from Millipore Sigma. Antibodies against dynein heavy chain (sc-514579; dilution 1:2000), dynactin p62 (sc-55604; dilution 1:2000), dynactin p150 (sc-135890; dilution 1:2000), dynactin p50 (sc-393389; dilution 1:2000) and dynactin Arp1 (sc-390632; dilution 1:2000) were obtained from Santa Cruz, Texas; antibody against dynein intermediate chain (D5167; dilution 1:5000) was obtained from Sigma-Aldrich; HRP-conjugated secondary antibody against mouse IgG (ab97023; dilution 1:20,000) was obtained from Abcam.

### Molecular cloning and baculovirus production
All plasmids were constructed by polymerase chain reaction and conventional restriction endonuclease and DNA ligation methods. Vector backbones with 3C protease cleavage sites and indicated fusion tags were provided by the protein expression, purification, and chromatography (PEPC) facility at the Max Planck Institute, Cell Biology and Genetics, Dresden, Germany[60]. cDNA encoding DIC2-SNAPf and p62 were introduced into a vector downstream of the human cytomegalovirus (CMV) immediate-early promoter and enhancer with C-terminal affinity tags- MBP for DIC2-SNAP and TwinStrep for p62. A gBlock sequence (Integrated DNA Technologies) coding for 8xHistidine tag and streptavidin binding protein (SBP) in tandem was fused to

the ORF corresponding to either BICD2N594 or BICD2N594-eGFP at the 3′ ends using High Fidelity assembly reaction (New England Biolabs) and introduced into an *E. coli* expression vector with an N-terminal MBP affinity tag. The gene coding for KIF16B was subcloned into an insect expression vector with and without a C-terminal e-GFP tag and an MBP affinity tag. All constructs contained a PreScission 3C protease site between the gene-of-interest and the affinity tags. All plasmids were verified by DNA sequencing.

Recombinant baculovirus for DIC2-SNAPf, p62 and KIF16B (with and without eGFP) were produced in Sf9 cells using the FlexiBAC system[60]. Briefly, plasmid vectors with the gene of interest were co-transfected with a replication-defective bacmid DNA into Sf9 cells. Homologous recombination between flanking sequences in both piece of DNA introduces the gene of interest into the viral genome and rescues viral replication and subsequent amplification. While recombinant viruses of P2 or P3 were used for DIC2-SNAPf and p62 (depending on the titer), P2 viruses were used for KIF16B. Viruses were either stored at 4 °C for 1 month or were supplemented with 10% (w/v) sucrose (in PBS; phosphate buffered saline, pH 7.3) and stored at −80 °C for no more than 6 months.

## Protein purification and concentration estimation

**Dynein and dynactin.** We exploited the ability of recombinant baculovirus to infect mammalian cells[61] to introduce affinity-tagged DIC2-SNAPf or p62 into HEK293F cells as bait for the dynein and dynactin complex, respectively. Genes for bait proteins in tandem with an affinity tag (Maltose binding protein, MBP for IC2 and Twin-Strep for p62) were introduced into large-scale suspension cultures of HEK293 via BacMam[61]. These bait proteins are incorporated into their respective complexes and can be fished out from cell lysates using affinity chromatography[13]. Peak expression of either bait was determined by performing a time course study on a small-scale suspension culture of HEK293F cells. The expression level for both dynein and dynactin was measured by running equal amount of cell lysate, sampled at 24 hr intervals, on a 4–12% BisTris SDS-PAGE precast gel in MOPS buffer (Life Technologies), which were then blotted onto a PVDF membrane (BioRad) and probed with either a primary anti-dynein intermediate chain or anti-dynactin p62 antibody and secondary anti-mouse IgG-HRP. Sufficient expression of p62 required the addition of sodium butyrate to a final concentration of 5 mM 6 h post infection. Sodium butyrate inhibits histone deacetylases which, otherwise, suppress protein expression from extrachromosomal DNA[62–64]. Peak expression of both DIC2-SNAPf and p62 was observed at 24 h post infection when infected with 1:100 and 1:50 P2 virus:culture (v/v), respectively.

To purify dynein or dynactin, 2–3 L of $2 \times 10^6$ cells/mL HEK293F cells were infected with the respective baculovirus and grown in suspension for 24 h at 37 °C and harvested by centrifugation. Cell pellets were washed with ice-cold PBS, resuspended in equal volumes of dynein lysis buffer (30 mM HEPES pH 7.4, 100 mM KCl, 2 mM MgCl2, 1 mM EGTA, 10% glycerol, 0.02% Triton X-100, 0.2 mM MgATP, 1 mM DTT, 2 mM PMSF and 1× protease inhibitor cocktail (cOmplete, Roche) and lysed with one passage through an ice-cooled Avestin Emulsiflex C-5 homogenizer at 3000–5000 psi. The lysate was clarified at $200,000 \times g$ for 60 min at 4 °C and the supernatant was then incubated with 2 mL of beads, equilibrated in dynein lysis buffer, for 120 min at 4 °C on a tube rotator. We used amylose resin (E8021S NEB) for dynein and streptactin beads (28-9355-99, Cytvia) for dynactin. Protein-bound beads were then collected with gravity flow 20 mL chromatography column (Econo-Pac, Bio-Rad), washed with 25 mL of dynein lysis buffer without PMSF and 25 mL of dynein elution buffer (30 mM HEPES pH 7.4, 148 mM potassium acetate, 2 mM magnesium acetate, 1 mM EGTA, 10% Glycerol, 0.02% Triton X-100, 0.2 mM MgATP, 1 mM DTT). Dynein complexes were released from the amylose resin by cleaving the MBP tag with 20 μg/mL of PreScission 3C-Protease (diluted in the dynein elution buffer, obtained from PEPC

facility, MPI-CBG) for 60 min at 4 °C. We also obtained fluorescently labeled dynein molecules by incubating the amylose resin-bound complexes with 5 μM Alexa647-SNAP ligand (NEB; prepared in dynein elution buffer) for 40 min at 4 °C, thoroughly washing out excess ligand with dynein elution buffer, followed by cleaving the complexes from the resin as mentioned before. Dynactin was eluted from the streptactin resin with 2 mL of 2.5 mM d-Desthiobiotin (prepared in dynein elution buffer). Complexes were concentrated to ~500 μL with a 100 K MWCO AmiconUltra (UFC8100, Amicon) centrifugal concentrator, filtered through a 0.22 μm cellulose acetate spin filter (Costar, Utah) and passed through a Superose 6 increase 10/300 GL size-exclusion column in dynein gel filtration buffer using a liquid chromatography system (ÄKTA pure). Peak fractions (corresponding to 11–13 mL retention volume) were pooled and concentrated to ~250 μL, aliquoted in volumes of 5 μL, snap-frozen and stored in liquid nitrogen. Dynein labeling was confirmed by running the 11–13 mL fractions (along with other appropriate fractions) on an SDS-PAGE gel and visualizing the fluorescent bands corresponding to Alexa647- IC2-SNAP upon excitation with a 647 LED lamp.

**BICD2N594.** *M. musculus* bicaudal 2 (BICD2) truncated to first 594 amino acids was expressed in *E. coli* pRARE cells transformed with plasmid DNA carrying the gene in 750 mL Luria–Bertani medium containing 50 μg/mL kanamycin. The cells were grown at 37 °C with continuous shaking at 180 rpm till the optical density of 0.6 at 600 nm. Protein expression was induced by adding isopropylthio-β-galactoside (IPTG) to a final concentration of 0.5 mM and shaking the culture at 180 rpm at 18 °C for 18 h. The cells were then harvested at $7500 \times g$ for 10 min at 18 °C and the pellets were resuspended in 35 mL BICD buffer (30 mM HEPES pH 7.4, 150 mM KCl, 2 mM MgCl2, 10% Glycerol, 0.02% Triton X-100, 1 mM DTT) containing 1× protease inhibitor cocktail (cOmplete, Roche) and lysed by passaging it three times through an ice-cold Emulsiflex homogenizer at 5000–10000 psi. The lysate was supplemented with 10 μL Benzonase and spun at $186,000 \times g$ at 4 °C for 45 min in a Type Ti-45 rotor (Beckman-Colter) and the supernatant was passed through a 0.45 μm filter. The clarified lysate was incubated with pre-equilibrated amylose resin (one wash with water and two washes with BICD buffer) for 2 h at 4 °C on a rotary mixer. The resin was collected in a fresh gravity flow column, washed with 50 mL BICD buffer, and eluted with 20 mM maltose (supplement in BICD buffer). Constructs were either directly processed for protease cleavage or subjected to nickel–sepharose affinity chromatography using the C-terminal 8xHis tag. Such tandem purification with affinity tags at the N and C termini allow the exclusion of truncated proteins from the final preparations. Eluted proteins were passed through a 1 mL HisTrap HP column (29051021, Cytiva), which was pre-equilibrated with 10 column volumes (CV) of BICD buffer containing 30 mM imidazole followed by a wash with 10 CV of 30 mM imidazole containing BICD buffer and elution with 300 mM imidazole containing BICD buffer. Eluted proteins were incubated with 5 μg/mL of PreScission 3C-protease at 4 °C overnight, concentrated using 30 K MWCO spin filters and subjected to size-exclusion chromatography on a Superose 6 increase 10/300 column. Peak fractions (12–14 mL retention volume) were collected, concentrated, aliquoted, snap-frozen in liquid nitrogen and stored at −80 °C.

**KIF16B.** KIF16B was expressed and purified from Sf9 using baculovirus. Optimal expression conditions (infection ratio of 1:100 virus: culture, v/v, for 72 h at 27 °C) were identified by performing a time-course experiment, similar to that described for DIC2-SNAPf and p62 above with one difference: protein expression in cell lysates was observed by Coomassie (LC6065, SafeStain) staining of SDS-PAGE gels. Fresh or snap-frozen cell pellets from 500 mL, $1 \times 10^6$ cells/mL of Sf9 suspension culture infected with 5 mL of P2 virus carrying the gene for KIF16B were resuspended in 25 mL equilibration buffer (25 mM HEPES pH 7.2, 1 M

NaCl, 5 mM MgCl₂, 5% Glycerol, 0.1 mM ATP, 0.25% 3-((3-cholamidopropyl) dimethylammonio)-1-propanesulfonate (CHAPS), 10 mM β-mercaptoethanol) supplemented with 1× protease inhibitor cocktail (cocktail III Merck calbiochem, 535140) and 1 μL Benzonase (1 mg/mL, MPI-CBG) to shear any DNA released during lysis. Cells were lysed by two passages through an ice-cold Avestin Emulsiflex C-5 homogenizer at 3000–5000 psi and spun at 186,000 × g for 60 min at 4 °C. The supernatant was filtered through a 0.45 μm syringe filter and incubated with 3 mL of pre-equilibrated amylose resin (one wash with distilled water and three washes with ice-cold lysis buffer, bead volume = 1.5 mL) for 2 h on a rotary mixer in the cold room. Beads were collected in an empty gravity flow column, washed twice with 10 mL equilibration buffer, and eluted with 3 mL of equilibration buffer supplemented with 20 mM maltose. Eluted proteins were diluted to 10 mL with equilibration buffer and incubated with 5 μg/mL of PreScission 3C-protease (Histagged) on a rotary mix overnight in the cold room. The protease was removed by incubating the mixture of protein and protease with 1 mL of pre-equilibrated Ni-NTA resin (30210, Qiagen) for 60 min followed by passage through a fresh gravity-flow column. The flow-through was concentrated in a 30 K MWCo spin filters to ~1 ml, and gel-filtered on a Superose 6 increase 10/300 column in equilibration buffer at 4 °C. Peak fractions were pooled, and concentrated using fresh 30 K MWCo spin filters, aliquoted, snap-frozen in liquid nitrogen and stored at −80 °C.

SDS-PAGE was used to determine the purity of isolated proteins, and western blotting was used to confirm the identity of dynein heavy chain and dynein intermediate chain in dynein preparations and dynactin p150, dynactin p62, dynactin p50 and dynactin arp1 in dynactin preparations. Protein estimation was performed by running serial dilutions of a standard 6xHis-eGFP construct (MPI-CBG) alongside appropriate dilutions of the proteins on a 4−12% BisTris SDS-PAGE precast gel in MOPS or MES buffer (LifeTechnologies), stained with Coomassie for 60 min and de-stained in distilled water. Stained gels were imaged in an imaging station (c300, Azure Biosystems). The integrated intensity of the protein bands of interested was quantified using the "Gels" tool in Fiji[65]. Individual lanes were defined using the rectangle region-of-interest tool, intensity profiles of all lanes were plotted, and integrated intensities were recorded by selecting the area under each peak. Linear fitting of the integrated intensity vs. concentration of 6xHis-eGFP provided a calibration curve which was then used to estimate the concentration of desired proteins. The molar concentrations of DHC and p150 were defined as the concentration of dynein and dynactin respectively.

**Tubulin.** Porcine brain tubulin was purified by two rounds of polymerization and depolymerization in high molarity PIPES[66]. Rhodamine labeled tubulin was prepared using the labeling kit from Invitrogen following manufacturer's instructions.

## Preparation of vesicles

All phospholipids were acquired as lyophilized powders and resuspended in chloroform except for PI3P where a 200:100:3 (v/v/v) mixture of chloroform:methanol:water was used as a solvent. A thin film of phospholipids mixed in the molar ratio mentioned in Supplementary Table 1 (with either of the two fluorescent markers Atto-647N or Atto-488) was deposited on the inner walls of a clean glass vial under a light and steady stream of nitrogen followed by drying in a vacuum for 4 h. Multilamellar vesicles (MLVs) were prepared by hydrating the lipid film in dynein buffer B (30 mM HEPES pH 7.4, 50 mM potassium acetate and 2 mM magnesium acetate) supplemented with 5% (w/v) sucrose to final lipid concentration of 1 mg/mL (1.28 mM) with vigorous shaking on a vortex and stored at −20 °C. MLVs with a total lipid mass of 50 μg were freeze-thawed five times followed by 10 cycles (21 passages) through an extruder (Avanti Polar) containing a phosphocellulose membrane with a pore size of 100 nm. Vesicles were stored on ice and used within 24−48 h of preparation. Size distribution of vesicles

(Supplementary Fig. 1) were measured in a dynamic light scattering instrument (Zetasizer, Malvern). The theoretical molar concentration of PI(3)P in the final vesicle solution was used as the molar concentration of vesicles.

## Preparation of polarity-marked microtubules

Polarity-marked microtubules were prepared by preferentially extending the plus-ends of short GMPCPP seeds in the presence of N-ethylmalemide modified tubulin (NEM-tubulin). Briefly, short bright seeds of 1:3 rhodamine labeled tubulin (final conc. 20 μM) were polymerized in the presence of 1 mM GMPCPP (NU-402, Jena Bioscience) in BRB80 for 15 min at room temperature. An extension mix comprising of 15 μM 1:9 rhodamine-labeled tubulin, 6 μM fresh NEM-tubulin (40 μM unlabeled tubulin incubated with 1 mM NEM on ice for 10 min and excess NEM quenched with 10 mM DTT for 10 min), 2 mM GTP and 2 mM MgCl₂ in BRB80 was assembled on ice, warmed at 37 °C for 1 min and incubated with 1/20th volume of bright seeds at 37 °C for 20 min. Microtubules were stabilized with 10 μM taxol (Paclitaxel, Sigma) and harvested by spinning at 17,000 × g for 15 min. Polarity-marked microtubules were used within 72 h.

## Motility experiments

Water-tight flow channels were prepared from silanized glass coverslips (22 × 22 mm and 18 × 18 mm; cleaned in piranha solution and treated with 0.05% dichlorodimethylsilane) by placing 1.5 mm parafilm strips on a 22 × 22 mm coverslip (~3 mm) apart, covering with an 18 ×18 mm coverslip, and melting the parafilm at 55 °C heat block. All solutions were prepared in dynein buffer B. The channels were sequentially perfused with solutions containing TetraSpeck microspheres (diameter 0.1 μm, Invitrogen) for drift and color correction, monoclonal anti-β-tubulin antibodies to immobilize microtubules and 1% Pluronic F127 to block the surface. After a 60 min incubation, channels were washed twice with dynein buffer B and once with dynein buffer B containing 10 μM taxol and incubated with polarity-marked microtubules for 5 min.

For single-molecule DDB motility, 50 nM dynein and 100 nM dynactin were first incubated on ice for 5 min (=50 nM DD). In total, 38 nM DD was then mixed with 75 nM BICD2N594-eGFP in dilution buffer (dynein buffer B supplemented with 20 mM glucose, 0.1 mg/mL casein, 2.5 mM MgATP and 1 mM DTT) and incubated for 5 min on ice (=38 nM DDB). DDB complexes were diluted 4-fold with the imaging buffer (dynein buffer B supplemented with 20 mM glucose, 0.1 mg/mL casein, 2.5 mM MgATP, 1 mM DTT, 100 μg/mL glucose oxidase, and 20 μg/mL catalase) and perfused into flow channels. Single molecule motility of KIF1B-eGFP was observed by serially diluting the stock solution of KIF16B-eGFP to a final concentration of 1 nM in imaging buffer and perfusing into the flow channel.

Dual-motor vesicle assays were performed by incubating 120 nM vesicles with 840 nM BICD2N594 and different concentrations of KIF6B (mentioned in the text) or 25 nM KIF16B-eGFP (to detect the presence of KIF16B on minus-end directed and reversing vesicles) for 2 min on ice. In total, 24 nM of these vesicles were then incubated with 38 nM DD for 5 min on ice, diluted 4-fold in imaging buffer, and perfused into the flow channels. Single-motor vesicle assays were performed by replacing either KIF16B (for DDB−vesicles) or DDB (for KIF16B−vesicles) from the above mixture with a dilution buffer.

## Data acquisition

TIRF Microscopy was performed on a Nikon Eclipse Ti2 microscope equipped with a perfect focus system and a 100 × , 1.49 NA oil, apochromat TIRF objective, and 1.5 × optovar. Samples were illuminated with either 488, 561, or 647 nm lasers (100 mW each) placed in a visitron laserbox and channeled through an iLas2 ring TIRF module operated in a single angle mode (point TIRF). Images from different fluorescent channels were acquired with separate EMCCD cameras

(one iXon Life EMCCD each for 488 nm and 561 nm channels and iXon Ultra EMCCD for 647 nm channel), each containing $1024 \times 1024$ pixel sensor and controlled with VisiView. The size of each pixel was $87 \times 87$ nm. Images were either acquired in streaming mode with 100 ms exposure (10.0 frames per second) or every 300 ms in time-lapse mode with 100 ms exposure (3.3 frames per second).

## Data processing and analysis

**Tracking.** Single molecules of DDB-eGFP and KIF16B-eGFP, and vesicles were tracked with Fluorescence Image Evaluation Software for Tracking and Analysis (FIESTA)[67], which automates Gaussian fitting (stretched Gaussian for quantifying elongations; symmetric Gaussian otherwise) of fluorescence signals to extract X−Y position coordinates. All tracks were manually curated to exclude erroneous tracks from further analysis. Tracks from vesicles at microtubule junctions and from vesicles that remained stationary throughout were discarded. Any tracks from vesicles colliding with each other and slowing down, as a result, were also excluded. Tracks were corrected for drift and chromatic aberration (wherever applicable) by using TetraSpeck microspheres (Invitrogen) as fiduciary markers. The X−Y position trace of individual tracks was used as a path to calculate displacement along the path. This was necessary to avoid false directional reversals when simple a distance to the origin is calculated for vesicles moving on curved microtubules. The orientation of the tracks was adjusted such that motion towards the plus-end of the microtubule reflected positive displacement.

**Segmentation.** The segmentation of trajectories into runs and pauses is generally a difficult task since the cargo velocity fluctuates stochastically. This implies that there is an intrinsic ambiguity in defining the transitions between different motility states. Here, we introduced a simple but efficient Monte Carlo method for this task.

An optimal segmentation is a piecewise linear approximation $f(t)$ of the cargo trajectory which captures all significant velocity changes. The linear segments are connected by change points, which have to be placed in a way that minimizes the error function.

$E_{\text{err}} = \sum_{i=1}^{N} (x_i - f(t_i))^2$ of the piecewise linear approximation $f(t)$ ($x_i$ denotes the position of the cargo at time $t_i$, N the number of measurement points).

A trivial solution, which minimizes $E_{\text{err}}$, would be to introduce a linear segment between all neighboring data points. This solution, however, would not distinguish between significant changes in the cargo velocity and short-term fluctuations. Therefore, we have to limit the number of change points. This can be done by introducing additional costs $\mu$ for introducing a change point. So, the total cost function $H$, which has to be optimized by our Monte Carlo approach reads as follows:

$$H = E_{\text{err}} + \mu N_{cp}$$

where $N_{cp}$ denotes the number of change points.

The function $f(t)$ is given by:

$$f(t) = x_k^{cp} + \frac{x_{k+1}^{cp} - x_k^{cp}}{t_{k+1}^{cp} - t_k^{cp}} (t - t_k^{cp})$$

where $x_k^{cp}, t_k^{cp}$ are the position and time coordinates of change point $k$ and $t_k^{cp} \leq t < t_{k+1}^{cp}$.

Now, in order to optimize the number and positions of the change points we are using the following algorithm:

**Initialization.** We place a change point at the beginning $x_0^{cp} = x_0, t_0^{cp} = t_0$ and at the end $x_{N_{cp}+1}^{cp} = x_N, t_{N_{cp}+1}^{cp} = t_N$ of the trajectory. These change points are fixed, in order to make sure that $f(t)$ spans the whole trajectory. The other $N_{cp}$ change points are initially placed in equal

temporal distance $\Delta t = \Delta T / (N_{cp} + 1)$ on the experimental track, where $\Delta T = t_N - t_0$ denotes the duration of the track under consideration. The coordinates $(x_k^{cp}, t_k^{cp})$ and the number $N_{cp}$ of inner change points ($k = 1, 2, \ldots, N_{cp}$) will be optimized in the following.

**Equilibration.** In order to optimize the segmentation we update the coordinates $(x_k^{cp}, t_k^{cp})$ according to: $x_{k,\text{new}}^{cp} = x_k^{cp} + \epsilon_x \times (u - 0.5), t_{k,\text{new}}^{cp} = t_k^{cp} + \epsilon_t \times (u - 0.5)$ where $u$ is a random number between zero and one, $\epsilon_x, \epsilon_t$ parametrize the amplitude of the steps in space and time direction. We also update the number of change points: $N_{cp,\text{new}} = N_{cp} \pm 1$ by deleting a randomly selected change point or adding a change point at a random position on the track. The updated segmentation is accepted with probability: $p = \min(1, \exp(-\beta \Delta H))$, where $\Delta H = H_{\text{new}} - H$ and $H_{\text{new}}$ is the value of $H$ for the updated configuration of change points. The distances between cargo positions and $f(t)$ are given in units of nm. Using this parametrization of the dimensionless distance, we obtained high acceptance rates for $\beta = 0.005$ and $\mu = 10,000$. In order to equilibrate the system we update the change point configuration 4000 times.

**Cooling.** In order to find a configuration of change points that represents a local minimum of $H$ we gradually increased $\beta$ every 1000 updates by one order of magnitude, starting with $\beta = 0.005$ ending with $\beta = 50$.

**Post-processing.** Having identified the change points, the track is segmented into pauses and positive/negative runs. A segment between two change points is assigned as a pause when the slope between the first and the last point of the segment is less than 100 nm/s. If a segment is just one data point long the single data point is added to the following segment. If the distance traveled during a segment assigned as run is less than 500 nm, the segment becomes a pause.

**Velocity, spatial pause frequency, pause duration.** Instantaneous velocity was defined as the ratio of change in displacement and time between two consecutive frames. The velocity of positive and negative runs was defined as the arithmetic mean of instantaneous velocities of individual runs.

Spatial pause frequencies, $x_f$, of individual tracks were calculated as $x_f^i = \frac{x_p^i}{x_d^i}$, where $x_p^i$ is the number of pauses and $x_d^i$ the absolute distance traveled (absolute value for minus runs) of an individual track $i$. The spatial pause frequency of individual tracks was weighted by the proportion of total distance traveled by vesicles of a given type (minus, plus or reversal tracks), $w_i = \frac{x_d^i}{\sum_{i=0}^{N} x_d^i}$, to calculate the weighted mean spatial pause frequency (or simply pause frequency, $\bar{x}_f$), weighted standard deviation ($\sigma_f$) and weighted standard error of mean ($\sigma_{\bar{x}}$).

$$\bar{x}_f = \sum_{i=0}^{N} w_i \cdot x_f^i, \quad \sigma_f = \sqrt{\frac{\sum_{i=0}^{N} w_i \left(x_f^i - \bar{x}_f\right)^2}{\left(\frac{N-1}{N}\right)}}, \quad \sigma_{\bar{x}} = \frac{\sigma_f}{\sqrt{N-1}}$$

$N$ is the total number of tracks. Pause duration was simply defined as the time interval between the beginning and end of a pause segment.

**Statistics.** Pairwise Mann–Whitney $U$, two-sample $t$-test, two-, and one-sample Kolmogorov–Smirnov tests were performed using the python module *scipy.stats*[68]. Bonferroni corrections of $p$-values were performed manually by multiplying the $p$-values given by the tests by the number of comparisons made. Two- and one-sample weighted Kolmogorov–Smirnov test was manually implemented based on the source code of the two-sample KS-test from *scipy.stats* (function '*ks_2samp*' in file *scipy/stats/stats.py*). Instead of the unweighted

empirical cumulative distribution function

$$F(x_i) = \sum_{j=1}^{i} 1/N = i/N$$

the weighted empirical cumulative distribution function

$$F^w(x_i) = \sum_{j=1}^{i} w_j$$

with weights $w_j$ was used. *P*-values for all tests were computed from two-tailed tests.

**Data representation.** The Beeswarm plot was generated using the python package *seaborn*[69]. All other plots were generated using the python package *matplotlib*[70] and formatted in Inkscape. Kymographs and timelapse images were created with Fiji[65].

## Mathematical modeling

The mathematical model used in this work is based on previously published models[26,42,55] and has been adapted to the experimental geometry. In the model, we assign a fixed number of motors to each cargo. It is expected that in the experiment, the number of motors as well as the composition of KIF16B and DDB motors, slightly fluctuates between cargoes for a given concentration. To mimic these fluctuations, the number of motors (including active, inactive, and diffusive DDB and KIF16B) is thrown from a Gaussian distribution, with the mean being the given mean number of motors. The standard deviation of the Gaussian is a function of the mean:

$$\sigma(\mu) = 1.0683\sqrt{\mu}$$

This relation was found in an extra simulation, where varying numbers of motors were randomly distributed over a fixed number of cargoes. For a given number of motors, the actual number of KIF16B and DDB is randomly generated with probabilities given by the mean numbers of DDB and KIF16B. Thus, besides the number of motors, the ratio of KIF16B to DDB can slightly vary between cargoes for the same given mean numbers of KIF16B and DDB.

The cargo is divided into an attachment area and a reservoir similar to previous published models[36,37]. Motors in the attachment area can attach the microtubule, while motors in the reservoir are too far away to interact with the microtubule. In our model, we only take the motors in the attachment area into account but let them instantaneously exchange with the motors in the reservoir. The high diffusivity of motors in the vesicle membrane (10 µm²/s[38]) allows us to assume that motors reach a uniform distribution over the cargo surface between two attachment events such that an exchange with a reservoir is a valid approach.

Consequently, each time a motor attaches, it is randomly chosen whether the motor is active, inactive, or diffusive (in the case of DDB, see later for the definitions of active, inactive, and diffusive motors). Furthermore, the motor is assigned an individual maximal stepping rate upon attachment (see later for the definition of motor stepping rates). However, the ratio between DDB and KIF16B, which was assigned previously to this particular cargo, is fixed.

The microtubule is modeled as a one-dimensional lattice (one-dimensional coordinate system) with seven parallel lanes, which represent the protofilaments accessible to the motors, i.e., the upper half of the microtubule. KIF16B and DDB motors, which are bound to the cargo, can bind to the microtubule, step on it, and detach from the microtubule. Both KIF16B and DDB motors bind to the microtubule with constant attachment rates $k_{a,KIF16B}$ and $k_{a,DDB}$, respectively (see Supplementary Tables 3 and 4 for parameter values and references). When attaching to the microtubule, a Gaussian

distribution, peaked around the central protofilament, is used to randomly choose the lane (protofilament) the motor interacts with. The Gaussian distribution is cut at $\pm 3\sigma$, where the standard deviation sigma is one and the mean is zero (Supplementary Fig. 11). Once attached to a lane, the motors stay on this lane until they detach again.

When bound to the microtubule, motors exert a force on the cargo proportional to the motor deflection:

$$\Delta x^i(t) = x_{mt}^i(t) - X_V(t)$$

where $x_{mt}^i(t)$ is the motor head position on the microtubule and $X_V(t)$ the cargo (vesicle) position in the one-dimensional coordinate system parallel to the protofilament axis. The deflection does not depend on the selected lane.

The motors are modeled as Hookean springs with non-zero rest length $L_{KIF16B}$ and $L_{DDB}$, respectively. This means the forces the motors exert on the cargo are given by

$$F^i(t) = \begin{cases} \kappa_{KIF16B}(\Delta x^i(t) - L_{KIF16B}), & \text{if } (\Delta x^i(t) > L_{KIF16B} \\ \kappa_{KIF16B}(\Delta x^i(t) + L_{KIF16B}), & \text{if } (\Delta x^i(t) < -L_{KIF16B} \\ 0, & else \end{cases}$$

for KIF16B and by

$$F^i(t) = \begin{cases} \kappa_{DDB}(\Delta x^i(t) - L_{DDB}), & \text{if } (\Delta x^i(t) > L_{DDB} \\ \kappa_{DDB}(\Delta x^i(t) + L_{DDB}), & \text{if } (\Delta x^i(t) < -L_{DDB} \\ 0, & else \end{cases}$$

for DDB. $\kappa_{KIF16B}$ and $\kappa_{DDB}$ are thereby the KIF16B and DDB specific motor stiffnesses, respectively.

For individual maximal stepping rates under zero load, experimentally measured single-molecule KIF16B and DDB instantaneous velocities were used (Fig. 1c of the main text). For stepping under load, the stepping rate depends on the force regime. If the motor experiences a resisting force ($F^i(t) > 0$ for KIF16B and $F^i(t) < 0$ for DDB) smaller than the stall force ($F_{s,KIF16B} > F^i(t) > 0$ for KIF16B and $-F_{s,DDB} < F^i(t) < 0$), the stepping rate depends on the force and the ATP concentration, as suggested by Schnitzer et al.[71]:

$$s([ATP], F^i) = \frac{V_{max} \times [ATP]}{[ATP] + K_M} = \frac{k_{cat}(F^i) \times [ATP]}{[ATP] + k_{cat}(F^i)/k_b(F^i)}$$

with $k_{cat}(F^i)$ and $k_b(F^i)$ being Boltzmann-like distributed:

$$k_m(F^i) = \frac{k_m^0}{p_m + q_m e^{F^i \delta/k_B T}} \quad \text{with } m \in \{cat, b\}$$

While $k_{cat}^0 = v_f/d$ is determined by the motor forward velocity $v_f$ and step size $d$, the parameters $k_b^0$, $p_b + q_b = 1$ and $p_{cat} + q_{cat} = 1$ are taken from Schnitzer et al.[71]. $\delta$ is determined by setting the stepping rate at the stall force equal $0.1\,s^{-1}$, i.e., $s(ATP, F_s^i) = 0.1\,s^{-1}$.

Under assisting forces ($F^i(t) < 0$ for KIF16B and $F^i(t) > 0$ for DDB), the stepping rate is equal to the force-dependent stepping rate at zero load: $s([ATP], F^i) = s([ATP], F^i = 0)$. If the motor experiences resisting forces beyond the stall force ($F_{s,KIF16B} < F^i(t)$ for KIF16B and $-F_{s,DDB} > F^i(t)$ for DDB), the motor steps backward at a small but constant rate $s = v_b/d$.

While the formulas for the stepping rates are the same for KIF16B and DDB, the parameters such as forward velocity $v_f$ and stall force $F_s$ are different for KIF16B and DDB. Different stall forces and forward velocities lead to different force and ATP dependences for KIF16B and DDB. Even though the used force and ATP-dependent stepping was found for kinesin-1, the previously found force dependence of KIF16B and DDB are similar[23,72].

For stepping and attachment, steric motor hindrances (exclusion effects) are taken into account on the protofilaments. This means a motor can neither attach nor step to an occupied spot on the same lane of the microtubule.

For KIF16B and DDB detachment, an exponentially-increasing detachment rate is used following previous work[25,44,73]:

$$k_d(F^i) = k_d^0 \, e^{-|F^i|/F_d}$$

Different force-free detachment rates and detachment forces were used for KIF16B and DDB, respectively. The used force-free detachment rates $k_d^0$, were taken from experimentally measured run length and velocities of single KIF16B and DDB molecules, respectively.

Moreover, in single-molecule experiments, we observed that ~20% of KIF16B motors do not step at all. Therefore, the model also includes 20% inactive KIF16B motors, which do not step when being attached to the microtubule but rather stay strongly bound. Thus, we assign them a lower force-free detachment rate than for the active motors. For DDB, it was found experimentally that about 10% do not step at all, and about 10% diffuse along the microtubule. Thus 10% of DDB motors are modeled as inactive motors with lower force-free detachment rate and 10% of DDB motors are modeled to diffuse in the harmonic potential of the motor spring with the following rate

$$s_\pm(F^i) = s_0 e^{\mp|F^i|\cdot d/2k_BT}$$

The diffusing motors hence always tend to step towards their equilibrium positions where they are not under tension.

The Gillespie stochastic simulation algorithm[74] is used to advance the system in time. The motion of the cargo is modeled in the over-damped limit, which is in agreement with experimental conditions. This means after each motor update, the cargo diffuses in the harmonic potential of the attached motor springs around its equilibrium position, i.e., the position where forces exerted on the cargo are equilibrated. For the cargo diffusion, the Metropolis algorithm is used[75]. The simulation starts with no motor being attached to the microtubule. The measurement begins after a relaxation time of 4 s. The simulation is terminated either when no motor is attached or after 80 s. A number of simulated cargoes are provided in respective figures. Model parameters were either obtained from experiments or from literature, if possible. Certain parameter values obtained from the literature as well as some unknown parameter values, were optimized in order to fit the simulation best to the experiment. We made sure that parameter values remained in the range of those given in the literature. All parameters for DDB and KIF16B used in this work are summarized in Supplementary Tables 4 and 5, respectively.

Cargo forces are calculated as the sum of the absolute value of the forces exerted by motors attached to the microtubule on the cargo:

$$F_{\text{cargo}} = \sum_{i \in \{MT-\text{attached motors}\}} F_i$$

Simulated tracks are typically longer than experimental tracks. While microtubules are infinitely long in simulations, microtubules in vitro are of a finite length, and as a result, many vesicles reach the end of the microtubule. To account for the same time duration distribution of simulated tracks as the experiment, the simulated tracks were resampled. Each simulated track was cut in equal pieces of time durations similar to individual experimental tracks of the analogous condition. Parameters (velocities, pause durations) computed for each piece were then weighted (weights $= \frac{1}{\text{Number of pieces in a given condition}}$) to match the experimental track durations. Resampled tracks were used to produce Fig. 5b, c, e, Supplementary Fig. 7b, c, Supplementary Fig. 8a, b, Supplementary Fig. 10b.

## Reporting summary

Further information on research design is available in the Nature Portfolio Reporting Summary linked to this article.

## Data availability

The authors declare that the data used to generate all plots in this study are available in the source data file. The experimental and simulated tracks are available from the corresponding author upon request. Source data are provided in this paper.

## Code availability

The C++, Python and MATLAB scripts for generating simulated tracks and processing (including segmentation) experimental and simulated tracks are available from the corresponding author upon reasonable request.

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

## Acknowledgements

We thank Samara Reck-Peterson (University of California San Diego) for providing the constructs for DIC2-SNAPf and p62-Halo, Marino Zerial (MPI-CBG, Dresden) for providing the constructs for KIF16B, Regis Lemaitre and the Protein Expression, Purification, and Chromatography Facility of MPI-CBG for providing cloning and expression vectors, generating baculovirus and for providing Sf9 and HEK293 cells, Jens Ehrig and the Molecular Imaging and Manipulation Facility of CMCB at TU Dresden for assistance with microscopy, Corina Bräuer for technical assistance, Felix Ruhnow for assistance with image analysis, Veikko Geyer for fruitful discussions, as well as Marino Zerial, Laura Meissner, Stefan Golfier, Jens Ehrig and Aditya Chhatre for comments on the manuscript. We acknowledge funding from the German Research Foundation (SFB1027), the German Federal Ministry of Education and Research (OptiZeD 03Z22E511), and the Boehringer Ingelheim Fonds (Ph.D. stipend to A. I. D'Souza).

## Author contributions

A.I.D., R.G., and S.D. conceptualized and designed the experimental research; G.A.M. and L.S. conceptualized theoretical modeling and simulations; A.I.D and R.G. generated the protein constructs and performed the experiments; G.A.M. performed the theoretical modeling and simulations, A.I.D., R.G., and G.A.M. analyzed the data; all authors discussed the data; A.I.D., R.G., G.A.M., and S.D. wrote the paper with comments by all authors.

## Funding

## Competing interests

The authors declare no competing interests.
