## [Peer Review File · Nature Communications]

Vesicles driven by dynein and kinesin exhibit directional reversals without regulatorsREVIEWER COMMENTS

Reviewer #1 (Remarks to the Author):

In this paper titled “Vesicles driven by dynein and kinesin exhibit directional reversals without external regulators” by D’Souza et al, the authors employ a reconstitution based method to simulate vesicle cargo transport by motor complexes of opposite polarity preference (e.g. dynein and kinesin). When only one type of motor is used, the expected directional transport along polarized microtubules is observed and typical parameters such as instantaneous velocity and pause frequency/duration are calculated. As mixtures of motors are titrated, the authors observe an increase in reversal events which are often preceded by a unique pause state which appears to occur as a result of a tug-of-war style mechanism between the plus- and minus-end directed motors. They perform computational simulations to explain these results, and produce a model that describes how motors diffusing along the vesicle surface enter a special zone permissible to microtubule attachment to compete for binding sites on the filament. When tuned properly, this model recapitulates all essential features of the data. Together, this is an impressive study and takes the field well beyond previous analyses of multi-motor competition which relied on non-physiological attachment strategies like plastic beads or DNA origami. I support publication of this article after the authors address several points as described below:

Critiques:

The observation of vesicle stretching during a reversal-track pause (Figure 4c) is one of the most compelling data points which suggest this is indeed a tug-of-war mode, where motors are exerting opposing forces and deforming the vesicle. I would imagine that during minus-track or plus-track pauses, this deformation is not observed? It would be good to quantify this in some way as additional proof that these reversal pauses are indeed a unique ‘tug-of-war’ style pause state that is fundamentally different than a typical pause. Perhaps calculating the average ellipticity of the vesicle during pauses could achieve this, as the data presented in Figure 4c clearly shows a striking difference between circular particle during motility and elongated elliptical particles during pausing/stretching. This would really bolster the findings to support a tug-of-war behavior during this special class of pause, as the current analysis is limited to the observation that pause frequency and duration both increase relative to unidirectional pauses (Fig 4a/b).

The interpretations of the data and simulations both rely on the idea that motors attached to the vesicle can rearrange along the surface rapidly due to their attachment to diffusible lipids. Once they move from the reservoir to the attachment area, they may engage with the microtubule and produce force/motion. In this reconstituted system, I would suspect this diffusion rate is quite fast due to the

simple lipid composition of the vesicle. However, in real cells, vesicles contain myriad components like proteins and more rigid molecules like cholesterol, and therefore the diffusion rate of attached motors would likely be significantly slower. How would the data change if the lipid bilayers in these vesicles were much less fluid-like? Is it possible to incorporate cholesterol into this system and repeat the dual-motor experiments (at just the 38nM DDB / 25nM KIF16B condition) to see how reversal/pause behavior and motility properties are affected? Also, I would imagine this diffusion rate into and out of the attachment zone can easily be modified in the simulations; how are the results affected here?

The authors should include several example vesicle trajectory outputs from their computational simulations either in the main figure 5 or as supplemental data. It would be useful as a point of comparison with the numerous experimental data trajectories (e.g. Fig 2b-d, Fig 3a), and would provide a more thorough presentation beyond the broader parameter analysis from Fig 5b-e.

Reviewer #2 (Remarks to the Author):

D'Souza, Grover, Monzon et al. use *in vitro* experiments and mathematical modeling to reconstitute bidirectional microtubule-based transport of vesicles without the need for cell regulators (other than the motor complexes themselves). This work addresses a long-standing debate about whether opposite-polarity motors engage in a tug of war or coordinated

(‘regulated’) mechanism when moving vesicles in cells. Previous *in vitro* efforts to reconstitute bidirectional motility have either used (1) Rigid DNA linkers/chassis (Derr et al., 2012; Belyy et al., 2016), (2) Purified vesicles from cells (Hendricks et al., 2010 and others) or (3) activating adaptor scaffolds such as Hooks, and TRAKs that can bind simultaneously to opposite-polarity motors (Kendrick et al., 2019; Siddiqui et al., 2019; Fenton et al., 2021; Canty et al., 2021 *BioRxiv*).

In this study, Dynein-dynactin-BicD2 (His-tag) and KIF16B (kinesin-3 containing a PH domain) are linked to unilamellar vesicles coated with DGS-NTA (for DDB binding) and PI3P (for KIF16B binding). Their results/conclusions are: (1) vesicles containing opposite-polarity motors engage in a tug-of-war that recapitulate the motion observed in cells. In their *in vitro* assay, vesicles undergo a combination of unidirectional runs, pauses and reversals (aka, ‘bidirectional movements’). Since all proteins in their assays are from purifications, the authors conclude that the bidirectional motility behavior observed can occur without external regulators (cellular factors). (2) Vesicle velocity was not influenced by the presence of opposite polarity motors. This was likely because there are only a few motors in the “attachment” zone capable of engaging with microtubules at any given time. Therefore, during directional runs, the opposite polarity motors are either not engaged with the microtubule or inactive. (3) Low numbers of engaged motors are critical, as exchange between any single motor will have drastic effects on the motility of the vesicle.

Overall, I believe this study is elegant in design and executed well. It addresses a critical question in the field by using a reconstituted system (as opposed to purified vesicles from cells) and PIP-coated vesicles (as opposed to rigid linkers). It has some good supporting evidence to back up the claim that bidirectional vesicle movement can be reconstituted “without external regulators.” However, the study should address the following criticisms, either as new experiments or textual changes:

(1) This study would benefit greatly by contextualizing the findings into the broader scientific history, as it relates to bidirectional motility of vesicles on microtubules. Here are specific suggestions:

a. The claim in line 150 as well as other places in the manuscript suggest that the behavior they observe *in vitro* resembles “native cargoes *in vivo*”. Which cargoes are these? Specific references to cargoes using KIF16B for motility would be helpful. Comparing those specific cargoes with the observed behaviors in this manuscript is important, since this is a major point.

b. In lines 47/48, the authors, mention the numerous papers showing that activating adaptors can activate both dynein and kinesin. It is important to mention that these same activating adaptors also act as bidirectional scaffolds capable of recruiting opposite-polarity motors and have been used in reconstituted systems to study bidirectionality *in vitro*. I realize that this is touched upon in the Discussion, but it would be useful to mention in the Introduction when discussing artificial assemblies of motors.

c. Along the same lines, the Introduction (starting at line 64) and Discussion (lines 342-343) ignore the Hendricks et al., 2010 data which concluded that purified phagosomes from cells are capable of bidirectional transport. In that paper, the claim is made that additional cytosolic factors are not necessary. Furthermore, that study built upon Muller et al., 2008 which used mathematical modeling to come to the same conclusion. Please explain and update.

d. The authors describe a “small attachment area” where vesicle-bound motors can engage with MTs. This was described by others including Jiang et al. 2019 (Hancock lab). Please provide appropriate context and references for this.

(2) Directly compare the *in vitro* findings with cellular findings.

a. In line 41 of the Introduction, the authors state that “the direction of cargo transport is biased towards the intended intracellular destination”. Based on the current findings, how do the authors think this is achieved in cells? Is it due to the run lengths of dynein being greater than kinesin (for retrogradely-destined cargoes?). A discussion point here would be very useful.

b. Please compare/contrast how linking DGS-NTA to BicD2 is similar/dissimilar to a cargo interaction with Rab6 and full-length BicD2 (capable of autoinhibition).

c. In Figure 4, the authors determined that non-reversal pauses are shorter in duration than reversal pauses. Is this the case in cells (*in situ*) with a relevant cargo type (i.e., one driven by KIF16B and/or BicD2)? Please provide experimental evidence.

(3) Questions related to the in vitro mechanism:

- a. Is dynein always present on positive-runs? Data similar to Extended Data Figure 3a should be collected with labeled DDB. Along these same lines, how can the authors be certain that dynein/dynactin are always present with bound BicD2 on unilamellar vesicles?
- b. Do the authors believe that there is not a slow velocity component during the “pausing” phase when both motors are engaged and active (similar to what was observed in Belyy et al., 2016)? The authors should quantify this ‘pausing’ phase to determine if slower velocity behaviors exist during the tug of war events.
- c. The authors use the term inactive motors a number of times throughout the manuscript. What % of KIF16B associated with vesicles are dead motors?
- d. How often did the vesicle elongation occur and how pronounced is this finding? The authors should quantify the in vitro deformation of vesicles and determine if vesicles are more pronounced in reversals vs. non-reversals after a pause. Bin the data if only larger-diameter vesicles can be used.

Reviewer #3 (Remarks to the Author):

The paper presents results on bidirectional transport by kinesin and dynein motors reconstituted on vesicles and shows that bidirectional movement is rapid while unidirectional, but interrupted by pauses and reversals. The direction reversals that can be modulated by the relative motor concentrations. A mathematical model is used to obtain a detailed picture of molecular details that cannot be observed experimentally.

This is an excellent paper that presents beautifully designed experiments and shows highly interesting results that will provide a new input into the long debate of whether molecular motors need regulation to show bidirectional transport or whether mechanical tug-of-war is sufficient. The key difference to previous experiments is that in this case, the motors are mobile on the cargo (as they likely are on cargos in cells), which appears to make a big difference in the transport properties. This is clearly an important contribution to the field and the combination with modeling is very fruitful here. I am clearly in favor of accepting the paper for publication.

This said, I also have a few comments that the authors should consider in a revision:

- 1) A hallmark of a tug-of-war with opposite-polarity motors pulling the cargo in both directions is the stretching the cargo. This is mentioned here as something that is seen occasionally. I think this result could be made more quantitative. moreover, if no extension of the cargo is seen during a pause, do the

authors interpret this as something different from a tug-of-war or as a tug-of-war that cannot be detected?

Related to this, the stretching of cargo was first described in a lesser-known paper than ref. 9 by Gennerich and Schild, Phys. Biol 2006.

2) Again related to point 1, I am wondering whether a classification of pauses beyond diffusive and stationary and reversing and non-reversing is possible. In particular, there must of tug-of-war situations that do not result in reversals, so in fig. 4b, similar events (tug-of-war situations) are included in reversal pauses and non-reversal pauses, but these show very different duration statistics.

3) The results of the modeling part are not so clear in the results section and the detailed picture of the trajectories is only shown in the discussion. Maybe these two parts could be connected better and the description on p. 11 be better connected to the figures. I find it hard to see these conclusions from the average numbers of motors shown in fig. 5d and would suggest to also show typical simulated trajectories with numbers of engaged motors indicated and/or distributions of the motor active numbers.

4) One point I did not get about the model: is exchange with the reservoir instantaneous or does this have a timescale/rate? This might make a difference when modulating the reattachment rate.

5) I assume that force-dependent unbinding of motors is still required in the model proposed here, but is supported by the diffusive exchange of motors with the reservoir to generate reversals and bidirectional motion. Is this correct? In that respect, the reservoir exchange may effectively have similarities with motor activation by regulation, as discussed by Munoz& Klumpp (J Chem Phys 2022). Effects of motor diffusion on cargo have also been modeled by Yadav and Kunwar (Phys Biol 2022)

We thank the reviewers for their careful and constructive review of our manuscript. Please find below our detailed point-to-point responses to the reviewers' comments and our actions taken. In addition, the key changes are also indicated by blue color in a submitted version of the manuscript for review only.

In particular, we (i) performed additional experiments and analyses (e.g. using labeled dynein motors and further analyzing vesicle stretching), (ii) developed and applied a novel segmentation algorithm, and (iii) extended the discussion of our results.

With regard to the development and application of the novel segmentation algorithm (which is one of the major reasons for taking this revision a substantial amount of time) we would like to mention up front the following: Prompted by all reviewers remarking on our finding of longer durations for reversal pauses vs. non-reversal pauses, we updated our segmentation algorithm with regard to even more precisely determining the pause durations. In our first submission, we used a segmentation algorithm based on a previously established mean-square displacement method. During revision, we realized that this method does not always precisely localize the change points (when applied to both, our experimental and simulated data) such (i) that instantaneous reversals were often segmented as pauses of up to two seconds and (ii) that the durations of reversal pauses were occasionally slightly overestimated as compared to the durations of non-reversal pauses. In our revision, we now introduce a simple, but efficient, Monte-Carlo based method for this task. The Monte Carlo method finds an optimal piecewise linear approximation of the whole cargo trajectory which captures all significant velocity changes. The linear segments are connected by change points, which are placed in a way that the error-function describing the difference between the original track and the linear approximation of the track is minimized. Thereby the number of change points are tuned by introducing a cost function (similar to a chemical potential in a grand-canonical ensemble in thermodynamics) for each change point. In dependence of the slope of the equal-velocity segments, the segments are assigned to be pauses or directional runs. When we apply this new algorithm to determine the pause durations in our experimental data, we still find that reversal pauses last slightly longer than non-reversal pauses. However, when we apply the algorithm to our simulated data, we do not recapitulate significantly different durations for reversal and non-reversal pauses. This indicates that our new algorithm does not contain a systematic bias towards longer reversal pauses. As for the now slightly different outcome of the pause durations from experiment and simulation, we do offer a hypothesis related to the enhanced rebinding of detached motors stabilizing the tug-of-war configurations in the main text.

We now describe our new algorithm in detail in the Materials and Methods and reassessed all data. Understandably, this led to changes in the directionality frequencies, velocities, and pause frequencies/durations. However, these changes were small and did not lead (besides the pause durations in our simulated data) to any changes in the findings/statements of our original submission. Rather, we hope that the new segmentation algorithm now added to our manuscript will be found useful by many others when evaluating similar traces.

Reviewer #1 (Remarks to the Author)

In this paper titled "Vesicles driven by dynein and kinesin exhibit directional reversals without external regulators" by D'Souza et al, the authors employ a reconstitution based method to simulate vesicle cargo transport by motor complexes of opposite polarity preference (e.g. dynein and kinesin). When only one type of motor is used, the expected directional transport along polarized microtubules is observed and typical parameters such as instantaneous velocity and pause frequency/duration are calculated. As mixtures of motors are titrated, the authors observe an increase in reversal events which are often preceded by a unique pause state which appears to occur as a result of a tug-of-war style mechanism between the plus- and minus-end directed motors. They perform computational simulations to explain these results, and produce a model that describes how motors diffusing along the vesicle surface enter a special zone permissible to microtubule attachment to compete for binding sites on the filament. When tuned properly,

this model recapitulates all essential features of the data. Together, this is an impressive study and takes the field well beyond previous analyses of multi-motor competition which relied on non-physiological attachment strategies like plastic beads or DNA origami. I support publication of this article after the authors address several points as described below:

Critiques: The observation of vesicle stretching during a reversal-track pause (Figure 4c) is one of the most compelling data points which suggest this is indeed a tug-of-war mode, where motors are exerting opposing forces and deforming the vesicle. I would imagine that during minus-track or plus-track pauses, this deformation is not observed? It would be good to quantify this in some way as additional proof that these reversal pauses are indeed a unique ‘tug-of-war’ style pause state that is fundamentally different than a typical pause. Perhaps calculating the average ellipticity of the vesicle during pauses could achieve this, as the data presented in Figure 4c clearly shows a striking difference between circular particle during motility and elongated elliptical particles during pausing/stretching. This would really bolster the findings to support a tug-of-war behavior during this special class of pause, as the current analysis is limited to the observation that pause frequency and duration both increase relative to unidirectional pauses (Fig 4a/b).

Response: We did follow the suggestion of the reviewer (similar to a suggestion by reviewer #2) to further quantify the vesicle deformations. We introduced the elongation index (ratio of the difference to the sum of long and short axis of a vesicle) and performed additional dual-motor assays with larger vesicles. We did observe that elongations were more pronounced during lower velocities (including the pausing states) and less pronounced during higher velocities. These findings are consistent with our modeling where we observe more instances of higher absolute forces during pauses than during runs. Therefore, we conclude that dynein and kinesin predominantly engage in a tug-of-war during periods associated to vesicle pausing. **We added the new experimental data and analyses on vesicle stretching in the Main Text, Fig. 4d and Supplementary Fig. S5).**

Please, see also our general explanation above about the new segmentation algorithm and its implications on determining the durations of reversal and non-reversal pauses from experimental and simulated data.

The interpretations of the data and simulations both rely on the idea that motors attached to the vesicle can rearrange along the surface rapidly due to their attachment to diffusible lipids. Once they move from the reservoir to the attachment area, they may engage with the microtubule and produce force/motion. In this reconstituted system, I would suspect this diffusion rate is quite fast due to the simple lipid composition of the vesicle. However, in real cells, vesicles contain myriad components like proteins and more rigid molecules like cholesterol, and therefore the diffusion rate of attached motors would likely be significantly slower. How would the data change if the lipid bilayers in these vesicles were much less fluid-like? Is it possible to incorporate cholesterol into this system and repeat the dual-motor experiments (at just the 38nM DDB / 25nM KIF16B condition) to see how reversal pause behavior and motility properties are affected?

Response: We agree that it would be exciting to validate our predictions experimentally. Unfortunately, we believe such experiments are not readily feasible for the following reasons: Firstly, to perform the experiment correctly, one would have to characterize the motility of vesicles synthesized from DOPC and DOPC + cholesterol mixtures bound with an equal number of motors. However, incubating the vesicles with identical concentrations of motors does not guarantee that the number of motors associated with DOPC and DOPC + cholesterol vesicles will be equal. Estimating the number of motors associated with vesicles is non-trivial and will require detailed characterization and validation. We are working towards such quantification in our system (following ideas for example from ¹) but do not feel in the position to incorporate this methodology into our current work. Secondly, the presence of cholesterol affects several properties of a lipid membrane beside the diffusion of integral and peripheral proteins. The increased bending rigidity of cholesterol-containing membranes²⁻⁴, besides others, will confound the effects caused by the change in protein diffusion.

We will strive to follow up the suggestion of the reviewer in future experiments.

Also, I would imagine this diffusion rate into and out of the attachment zone can easily be modified in the simulations; how are the results affected here?

Response: In our model we consider a point-like cargo, which means that motor diffusion is not directly modelled. Instead, we use a reservoir, into which a motor goes after detachment and from where motors enter the attachment area before they attach to the microtubule. Because the experimentally given diffusion constant on a DOPC vesicle surface is (very) high ($10 \mu\text{m}^2/\text{s}$,⁵) it can be assumed that motors quickly reach a uniform distribution after any detachment event, i.e. there is no memory when a new motor (randomly drawn out of the reservoir) attaches to the microtubule. Having a significantly lower diffusion constant of motors on the vesicle surface, would have the following consequences: (i) A particular motor would not necessarily exchange with the reservoir but likely reattach to the microtubule (memory effect). Consequently, fluctuations in the microtubule-bound motor configuration (i.e. the composition of motors attached to the microtubule) would be reduced. (ii) Detached motors would not diffuse far away from the microtubule, but stay in close proximity. They will then quickly attach again leading to higher attachment rates. Along these lines, we find a reduction in the number of reversal tracks when we increase the attachment rate in our simulations (see Fig. 5e). Moreover, our simulations show that the number of reversal tracks also depends on the number of available detached motors in the attachment area, the number of attached motors, the motor properties (detachment and attachment rates) and the space for attachment available on the microtubule.

We now provide an expanded description of our model in the Materials & Methods clarifying why an exchange with the reservoir is a valid approach. Furthermore, we now explain in the Discussion what we expect to happen at lower diffusion rates.

The authors should include several example vesicle trajectory outputs from their computational simulations either in the main figure 5 or as supplemental data. It would be useful as a point of comparison with the numerous experimental data trajectories (e.g. Fig 2b-d, Fig 3a), and would provide a more thorough presentation beyond the broader parameter analysis from Fig5b-e.

Response: We apologize for not clearly stating that example tracks from the simulation were provided in the Supplementary Data. **We now clearly mention that several example tracks of simulated cargoes can be found in Supplementary Fig. S7 (same presentation as the experimental data).**

Reviewer #2 (Remarks to the Author)

D'Souza, Grover, Monzon et al. use in vitro experiments and mathematical modeling to reconstitute bidirectional microtubule-based transport of vesicles without the need for cell regulators (other than the motor complexes themselves). This work addresses a long-standing debate about whether opposite-polarity motors engage in a tug of war or coordinated ('regulated') mechanism when moving vesicles in cells. Previous in vitro efforts to reconstitute bidirectional motility have either used (1) Rigid DNA linkers/chassis (Derr et al., 2012; Belyy et al., 2016), (2) Purified vesicles from cells (Hendricks et al., 2010 and others) or (3) activating adaptor scaffolds such as Hooks, and TRAKs that can bind simultaneously to opposite-polarity motors (Kendrick et al., 2019; Siddiqui et al., 2019; Fenton et al., 2021; Canty et al., 2021 BioRxiv).

In this study, Dynein-dynactin-BicD2 (His-tag) and KIF16B (kinesin-3 containing a PH domain) are linked to unilamellar vesicles coated with DGS-NTA (for DDB binding) and PI3P (for KIF16B binding). Their results/conclusions are: (1) vesicles containing opposite-polarity motors engage in a tug-of war that recapitulate the motion observed in cells. In their in vitro assay, vesicles undergo a combination of unidirectional runs, pauses and reversals (aka, 'bidirectional movements'). Since all proteins in their assays are from purifications, the authors conclude that

the bidirectional motility behavior observed can occur without external regulators (cellular factors). (2) Vesicle velocity was not influenced by the presence of opposite polarity motors. This was likely because there are only a few motors in the “attachment” zone capable of engaging with microtubules at any given time. Therefore, during directional runs, the opposite polarity motors are either not engaged with the microtubule or inactive. (3) Low numbers of engaged motors are critical, as exchange between any single motor will have drastic effects on the motility of the vesicle.

Overall, I believe this study is elegant in design and executed well. It addresses a critical question in the field by using a reconstituted system (as opposed to purified vesicles from cells) and PIP-coated vesicles (as opposed to rigid linkers). It has some good supporting evidence to back up the claim that bidirectional vesicle movement can be reconstituted “without external regulators.” However, the study should address the following criticisms, either as new experiments or textual changes:

(1) This study would benefit greatly by contextualizing the findings into the broader scientific history, as it relates to bidirectional motility of vesicles on microtubules. Here are specific suggestions:

a. The claim in line 150 as well as other places in the manuscript suggest that the behavior they observe in vitro resembles “native cargoes in vivo”. Which cargoes are these? Specific references to cargoes using KIF16B for motility would be helpful. Comparing those specific cargos with the observed behaviors in this manuscript is important, since this is a major point.

Response: We have now included additional specific examples and references of native cargoes undergoing bidirectional transport whose motion is recapitulated in our in vitro assay.

b. In lines 47/48, the authors, mention the numerous papers showing that activating adaptors can activate both dynein and kinesin. It is important to mention that these same activating adaptors also act as bidirectional scaffolds capable of recruiting opposite-polarity motors and have been used in reconstituted systems to study bidirectionality in vitro. I realize that this is touched upon in the Discussion, but it would be useful to mention in the Introduction when discussing artificial assemblies of motors.

Response: We have updated the introduction to mention the scaffolding nature of certain cargo adaptors. This fits well with our intention of discussing how the directionality of a bidirectionally moving cargo can be biased (see also our revised Discussion).

c. Along the same lines, the Introduction (starting at line 64) and Discussion (lines 342-343) ignore the Hendricks et al., 2010 data which concluded that purified phagosomes from cells are capable of bidirectional transport. In that paper, the claim is made that additional cytosolic factors are not necessary. Furthermore, that study built upon Muller et al., 2008 which used mathematical modeling to come to the same conclusion. Please explain and update.

Response: We apologize for not citing the work of Hendricks et al. when claiming external regulators were not necessary to recapitulate the motility of native cargoes. This also made us realize that we had not properly defined external regulators. We consider anything that is not necessary for motility as external regulators. This would include soluble factors present in the cytosol, factors that associate with motors, and factors that might remain bound to the membrane. While the work of Hendricks et al. showed that cytosolic factors were not necessary, their use of native organelles (latex-bead-containing phagosomes) did not rule out the potential role of motor-associated or membrane-bound factors. In this regard, we believe our work goes one step further, showing that these factors are, indeed, not necessary to induce directional reversals.

We realize that by using the phrase ‘external regulators’ we might have lead readers to believe that we are referring to cytosolic regulators. Therefore, we dropped the word ‘external’ from

the title, text and revised our Introduction to explain what we mean by regulators, also in the context of the mentioned citations.

d. The authors describe a “small attachment area” where vesicle-bound motors can engage with microtubules. This was described by others including Jiang et al. 2019 (Hancock lab). Please provide appropriate context and references for this.

Response: We have now cited two works: Jiang et al 2019⁶ and Rai et al 2016⁷ when discussing the attachment area in our mathematical model (Main Text as well as Materials and Methods).

(2) Directly compare the *in vitro* findings with cellular findings.

a. In line 41 of the Introduction, the authors state that “the direction of cargo transport is biased towards the intended intracellular destination”. Based on the current findings, how do the authors think this is achieved in cells? Is it due to the run lengths of dynein being greater than kinesin (for retrogradely-destined cargos?). A discussion point here would be very useful.

Response: Yes, directionality is achieved when the overall lengths of the plus runs and minus runs are not balanced. The length of these individual runs will be determined by the configuration of motors on the cargo (i.e. number of active kinesin vs. active dynein motors). In addition, *in vivo*, it is expected that these motor configurations themselves, as well as the relative strengths of the individual motors, can be influenced by external factors such as adapters, MAPs and tubulin PTMs. **We now discuss this point in detail in the Discussion.**

b. Please compare/contrast how linking DGS-NTA to BicD2 is similar/dissimilar to a cargo interaction with Rab6 and full-length BicD2 (capable of autoinhibition).

Response: While our attachment scheme for dynein via truncated BicD2 and DGS-NTA is partially artificial, it emulates dynein recruitment to endogenous cargoes via cargo adaptor complexes. The truncated BicD2-DGS-NTA state might be similar to full-length BicD2-Rab6 complexes when the autoinhibition of BicD2 is relieved upon binding to Rab6⁸. **We now included an explanation about this resemblance into our Discussion.**

c. In Figure 4, the authors determined that non-reversal pauses are shorter in duration than reversal pauses. Is this the case in cells (in situ) with a relevant cargo type (i.e., one driven by KIF16B and/or BicD2)? Please provide experimental evidence.

Response: While we would like to corroborate our findings pertaining to pause frequency and duration inside cells, this undertaking will be non-trivial for the simple reason that there are causes other than the dynamics of opposite-motor coupling that will lead to cargo pausing. Many of these are beyond the control of the experimenter: microtubule crowding, obstacles, microtubule defects, etc. These issues are partly the reason why we are interested in taking an *in vitro* approach to bidirectional transport. **For the reasons given above, we would like to refrain from considering a comparison of our pause durations to existing cellular data.**

Please, see also our general explanation above about the new segmentation algorithm and its implications on determining the durations of reversal and non-reversal pauses from experimental and simulated data.

(3) Questions related to the *in vitro* mechanism:

a. Is dynein always present on positive-runs? Data similar to Extended Data Figure 3a should be collected with labeled DDB.

Response: To address this concern, we performed new experiments with labelled dynein molecules (AlexaFluor647-DDB) and unlabeled KIF16B, bound to Atto488 labelled vesicles. We observe that labelled dynein is indeed present on vesicles moving exclusively towards the plus-end as well as on vesicles exhibiting directional reversals. **We have included the description**

of these results in the Main Text and show the representative kymographs in Supplementary Fig. S3b.

Along these same lines, how can the authors be certain that dynein/dynactin are always present with bound BicD2 on unilamellar vesicles?

Response: We are certain that minus-end directed motility of vesicles is mediated by BicD2 bound dynein/dynactin, as incubation of vesicles with dynein/dynactin in the absence of BicD2 results in no motility.

b. Do the authors believe that there is not a slow velocity component during the “pausing” phase when both motors are engaged and active (similar to what was observed in Belyy et al., 2016)? The authors should quantify this ‘pausing’ phase to determine if slower velocity behaviors exist during the tug of war events.

Response: We segmented our traces into runs and pauses. The pauses can be diffusive, completely stationary or, occasionally, contain a slow-moving component, with velocities significantly lower than runs. However, in contrast to Belyy et al 2016, where completely stationary or slow-moving cargos in the presence of opposing motors were shown, the slow-motion states we observe eventually transitions into fast runs. **We now specify the pauses in more detail and show a box plot of segment velocities during the pauses (see new Supplementary Fig. S4).**

c. The authors use the term inactive motors a number of times throughout the manuscript. What % of KIF16B associated with vesicles are dead motors?

Response: We estimate that approximately 20% of KIF16B motors are inactive in our preparations from single molecule motility assays. **This information is now provided in the Materials and Methods (together with an approximation of inactive and diffusive DDB motors).**

d. How often did the vesicle elongation occur and how pronounced is this finding? The authors should quantify the in vitro deformation of vesicles and determine if vesicles are more pronounced in reversals vs. non-reversals after a pause. Bin the data if only larger-diameter vesicles can be used.

Response: We did follow the suggestion of the reviewer (similar to a suggestion by reviewer #1) to further quantify the vesicle deformations. We introduced the elongation index (ratio of the difference to the sum of long and short axis of a vesicle) and performed additional dual-motor assays with larger vesicles. We did observe that elongations were more pronounced during lower velocities (including the pausing states) and less pronounced during higher velocities. These findings are consistent with our modeling where we observe more instances of higher absolute forces during pauses than during runs. Therefore, we conclude that dynein and kinesin predominantly engage in a tug-of-war during periods associated to vesicle pausing. **We added the new experimental data and analyses on vesicle stretching in the Main Text, Fig. 4d and Supplementary Fig. S5).**

Reviewer #3 (Remarks to the Author):

The paper presents results on bidirectional transport by kinesin and dynein motors reconstituted on vesicles and shows that bidirectional movement is rapid while unidirectional, but interrupted by pauses and reversals. The direction reversals that can be modulated by the relative motor concentrations. A mathematical model is used to obtain a detailed picture of molecular details that cannot be observed experimentally.

This is an excellent paper that presents beautifully designed experiments and shows highly interesting results that will provide a new input into the long debate of whether molecular motors

need regulation to show bidirectional transport or whether mechanical tug-of-war is sufficient. The key difference to previous experiments is that in this case, the motors are mobile on the cargo (as they likely are on cargos in cells), which appears to make a big difference in the transport properties. This is clearly an important contribution to the field and the combination with modeling is very fruitful here. I am clearly in favor of accepting the paper for publication.

This said, I also have a few comments that the authors should consider in a revision:

1) A hallmark of a tug-of-war with opposite-polarity motors pulling the cargo in both directions is the stretching the cargo. This is mentioned here as something that is seen occasionally. I think this result could be made more quantitative. moreover, if no extension of the cargo is seen during a pause, do the authors interpret this as something different from a tug-of-war or as a tug-of-war that cannot be detected?

Related to this, the stretching of cargo was first described in a lesser-known paper than ref. 9 by Gennerich and Schild, Phys. Biol 2006.

Response: We added new experimental data and analyses on vesicle stretching (please see also our responses to the reviewer #1 (first comment) and reviewer #2 (last comment). We now also cite Gennerich and Schild 2006, when describing the stretching of the vesicle.

Experimentally, it is unfortunately not easy to observe vesicle stretching for elongations significantly shorter than the optical resolution of the imaging method (about 250 nm in our case). That is why we think that not observing stretching does not strictly mean that there is no tug-of-war.

2) Again related to point 1, I am wondering whether a classification of pauses beyond diffusive and stationary and reversing and non-reversing is possible. In particular, there must be tug-of-war situations that do not result in reversals, so in fig. 4b, similar events (tug-of-war situations) are included in reversal pauses and non-reversal pauses, but these show very different duration statistics.

Response: Yes, there are tug-of-war situations both during reversal and non-reversal pauses. Nevertheless, there are reasons for differences in the durations of reversal vs. non-reversal pauses. For example, part of the non-reversal pauses (but not the reversal pauses) consist of the binding of inactive motors only (i.e. without involving a tug-of-war between DDB and KIF16B). Such pauses are short (as we know from the pauses observed from single-motor vesicles) and the presence of these short pauses reduces the median duration compared to the reversal pause. This difference may be further enhanced by an increased rebinding probability of detached motors in the experiments (not considered in the model).

We now elaborate more clearly on the origin of pauses and the difference between the durations of reversal and non-reversal pauses.

Please, see also our general explanation above about the new segmentation algorithm and its implications on determining the durations of reversal and non-reversal pauses from experimental and simulated data.

3) The results of the modeling part are not so clear in the results section and the detailed picture of the trajectories is only shown in the discussion. Maybe these two parts could be connected better and the description on p. 11 be better connected to the figures. I find it hard to see these conclusions from the average numbers of motors shown in fig. 5d and would suggest to also show typical simulated trajectories with numbers of engaged motors indicated and/or distributions of the motor active numbers.

Response: We apologize for not being so clear in the modelling part. When describing the simulated tracks in the results, we now added a reference to the trajectories shown in the Supplementary Information. Moreover, we now better reference the adequate figures to

underline our conclusions and show time-distance plots including the number and kind of attached motors (Supplementary Fig. S9).

4) One point I did not get about the model: is exchange with the reservoir instantaneous or does this have a timescale/rate? This might make a difference when modulating the reattachment rate.

Response: We do model the exchange with the reservoir instantaneously because it is valid to assume that detached motors reach uniform distributions between two attachment events (typical diffusion distances are approx. 2000 – 2800 nm, while the diameter of the attachment area is about 160 nm). With 32-fold higher attachment rate, our approach would still be valid (typical distances travelled are approx. 500 nm). Such uniform distribution cannot be assumed anymore, for diffusion constants $D < 0.07 \mu\text{m}^2/\text{s}$ where the typical diffusion distance between two attachment events is in the same order of magnitude as the diameter of the attachment area. Please also see our answers to reviewer 1 for what we expect at low diffusion rates.

To make sure that our model is clearly described, **we have updated the model description in the Main Text as follows: “Upon detachment from the microtubule, motors are instantaneously exchanged for new motors ...”. Additionally, we improved the detailed model description in the Materials and Methods, where we now clarify that the exchange is instantaneous, and why this is a valid approach.**

5) I assume that force-dependent unbinding of motors is still required in the model proposed here, but is supported by the diffusive exchange of motors with the reservoir to generate reversals and bidirectional motion. Is this correct? In that respect, the reservoir exchange may effectively have similarities with motor activation by regulation, as discussed by Munoz & Klumpp (J Chem Phys 2022). Effects of motor diffusion on cargo have also been modeled by Yadav and Kunwar (Phys Biol 2022)

Response: Yes, our model includes force-dependent unbinding and motor exchange with the reservoir, which lead to varying attached motor configurations and consequently to reversals and bidirectional motion.

Our simulations show that the reservoir increases the percentage of reversal tracks, but reversals can also be seen without a reservoir as it has been shown by previous models (Mueller et al., Kunwar et al.). Importantly, however, we find that the attachment and force-dependent detachment rates are in a range such that a low number of motors is attached, i.e. (i) there is space on the microtubule for other motors to attach and (ii) a pool of unattached motors exists. Consequently, stochastic binding and force-dependent unbinding is sufficient to change the microtubule-bound motor configurations such that reversals occur.

We agree, another way to change the motor configuration would be a differential motor activation by regulators as it likely occurs *in vivo* and has been modelled by Munoz & Klumpp. However, we used our *in vitro* assay and computational simulations to study whether bidirectional transport, as observed *in vivo*, can be explained by a simple mechanical model without additional regulation. We find that biochemical regulation is not needed to see bidirectional motion, including reversals and long unidirectional runs.

To better describe our results, we now cite Munoz & Klumpp 2022 when discussing previous models who observed reversals and added a paragraph discussing previous modelling results. We also cite Yadav and Kunwar 2022.

References

1. Nelson, S. R., Trybus, K. M. & Warshaw, D. M. Motor coupling through lipid membranes enhances transport velocities for ensembles of myosin Va. **111**, E3986–E3995 (2014).
2. Chakraborty, S. *et al.* How cholesterol stiffens unsaturated lipid membranes. *Proc. Natl. Acad. Sci.* **117**, 21896–21905 (2020).

3. Doole, F. T., Kumarage, T., Ashkar, R. & Brown, M. F. Cholesterol Stiffening of Lipid Membranes. *J. Membr. Biol.* **255**, 385–405 (2022).
4. Subczynski, W. K., Pasenkiewicz-Gierula, M., Widomska, J., Mainali, L. & Raguz, M. High Cholesterol/Low Cholesterol: Effects in Biological Membranes: A Review. *Cell Biochem. Biophys.* **75**, 369–385 (2017).
5. Chiantia, S., Schwille, P., Klymchenko, A. S. & London, E. Asymmetric GUVs Prepared by M β CD-Mediated Lipid Exchange: An FCS Study. *Biophys. J.* **100**, L1–L3 (2011).
6. Jiang, R. *et al.* Microtubule binding kinetics of membrane-bound kinesin-1 predicts high motor copy numbers on intracellular cargo. *Proc. Natl. Acad. Sci.* **116**, 26564–26570 (2019).
7. Rai, A. *et al.* Dynein clusters into lipid microdomains on phagosomes to drive rapid transport toward lysosomes. *Cell* **164**, 722–734 (2016).
8. Huynh, W. & Vale, R. D. Disease-associated mutations in human BICD2 hyperactivate motility of dynein-dynactin. *J. Cell Biol.* **216**, 3051–3060 (2017).

REVIEWERS' COMMENTS

Reviewer #1 (Remarks to the Author):

The authors have done an excellent job addressing the concerns I, and the other reviewers, raised during the initial review process. At this point, with the additional data, analyses, and improvements to the text, I fully support publication of the manuscript with no additional edits required.

Reviewer #2 (Remarks to the Author):

D'Souza, Grover, Monzon et al. use in vitro experiments and mathematical modeling to reconstitute bidirectional microtubule-based transport of vesicles without the need for cell regulators (other than the motor complexes themselves). They address a long-standing debate and critical question in the field about whether opposite-polarity motors engage in a tug of war or coordinated ('regulated') mechanism when moving vesicles in cells.

As I mentioned in the first review, this study is elegant in design and executed well. They more than adequately addressed my concerns from the first revision.

Reviewer #3 (Remarks to the Author):

The revised version of the manuscript addresses all points that were raised. The new figures S5 and S9 are particularly appreciated and the new segmentation algorithm is an added bonus. I recommend to accept the paper in its present form.